# MOUSEDTB: A MOUSE DIGITAL TWIN BRAIN AT SINGLE-NEURON RESOLUTION

## ABSTRACT

Accurate whole-brain computational modeling grounded in single-neuron resolution connectivity is crucial for understanding how large-scale brain structures give rise to complex behaviors and cognition. Conventional mouse whole-brain models are typically constructed from coarse-grained regional or voxel-level connectivity, without considering single-neuron biological plausibility in the mouse brain connectome. In this study, we build a mouse digital twin brain (mouse DTB) at single-neuron resolution with large-scale spiking neural network, able to support complex behavioral tasks at whole-brain scale. We developed the mouse brain connectivity at single-neuron resolution through a data-driven pipeline that integrates high-resolution axonal projection data and spatial distributions of cells from the mouse brain cell atlas. The resulting neuronal connectivity is coupled with leaky integrate-and-fire (LIF) neurons and conductance-based synapses to form a large-scale spiking neural network of the mouse brain. The mouse DTB successfully reproduced blood-oxygen-level-dependent (BOLD) signals observed in both resting state and olfactory Go/No-Go discrimination task with high correlation, and exhibits correct behavioral responses aligned with perceptual odor inputs. This model leverages diffusion ensemble Kalman filtering (EnKF) and hierarchical Bayesian inference for parameter estimation. Our work provides a single-neuron resolution, whole-brain mouse DTB, offering a powerful tool for studying neural dynamics, behavior and cognition underlying mouse intelligence during complex tasks.

## 1 INTRODUCTION

Brain connectivity plays a crucial role in understanding how neural circuits give rise to behavior, cognition, and complex brain functions (Cook et al., 2019; Dorkenwald et al., 2024). Computational models of brain connectivity have been served as powerful tools for simulating how these structural networks support dynamic brain activity, offering insights into the mechanisms underlying perception, decision-making, and learning (Zhao et al., 2024; Shiu et al., 2024). However, current knowledge of mouse brain connectivity are constrained to either a detailed description of specific circuits (Glickfeld et al., 2013; Kleinfeld et al., 2011; Seeman et al., 2018; Lefort et al., 2009) and localized tissue (Tavakoli et al., 2025) or a coarse description of connectivity between larger brain regions (Oh et al., 2014) or voxels (Knox et al., 2018), which fails to capture the fine-grained long-range connectivity between individual neurons across the brain. Recent advances in tracing techniques and image processing have made it increasingly feasible to construct more refined and comprehensive connectivity (Oh et al., 2014), enabling the development of whole-brain computational models for investigating how network structures support complex cognitive processes and brain functions (Lu et al., 2024b;a).

Previous whole-brain models of mouse rely on coarse-grained connectivity at the scale of brain region or 100 $\mu$m voxel inferred from viral tracer experiments (Oh et al., 2014; Knox et al., 2018). Despite the mesoscale models have enhanced spatial resolution in connectivity and enabled prediction of brain-wide projection patterns, they are based on strong assumptions of homogeneity within voxels and fail to capture the variability of connectivity at the single-neuron level. Other efforts have focused on modeling specific isolated circuits, enabling detailed investigations of local neural dynamics (Billeh et al., 2020; Geiller et al., 2022; Ausborn et al., 2019; Oldenburg et al., 2024; Galván Fraile et al., 2024), but are inherently limited in revealing how distributed circuits across

the brain interact to support complex intelligent behaviors involving perception, decision-making, memory and motor control. As a result, these models are insufficient for uncovering the causal relationships between structural connectivity and emergent brain-wide dynamics that underlie complex brain functions.

To address these gaps, we aim to develop a digital mouse brain at single-neuron resolution based on high-resolution structural data and large-scale computational modeling. Our objective includes: (1) to infer a whole-brain neuronal connectivity based on 10 $\mu$m voxel resolution axonal projections and cell atlas data, and (2) to simulate brain dynamics in resting state and in action to validate the model's structural and functional realism. This effort opens the door to studying how brain-wide neuronal interactions support intelligent behaviors.

In this work, we proposed a data-driven pipeline to build a digital mouse brain model at single-neuron resolution from high-resolution structural data and functional magnetic resonance imaging (fMRI) signals in resting state and in action (Han et al., 2019), as show in Figure 1. First, 10 $\mu$m voxel-level connectivity are derived from axonal projections by employing kernel regression to approximate the projection weight of a given voxel as the distance-weighted sum of nearby injections (Oh et al., 2014; Knox et al., 2018). Neuronal density of each 10 $\mu$m voxel is estimated based on the spatial distribution of neurons from cell atlas (Zhang et al., 2023; Wang et al., 2020), and voxels are incorporated into individual neurons in breadth-first search (BFS) ordering, which enables the inference of neuronal connectivity from voxel-level connectivity. Secondly, the inferred neuronal connectivity are iteratively optimized to achieve a balanced distribution of neuronal out-degrees across different regions. Thirdly, Gaussian local connectivity are incorporated into the network (Potjans & Diesmann, 2014; Schmidt et al., 2018; Campagnola et al., 2022) to compensate for the loss of local connections within injection sites in projection data. Finally, we developed a large-scale spiking neuronal network with the resulting mouse brain neuronal connectivity using leaky integrate-and-fire (LIF) neurons and conductance-based synapses on the Digital Brain (DB) platform (Lu et al., 2024a;b). To estimate its parameters and align simulated activity with empirical observations, we employed a data assimilation framework combining diffusion ensemble Kalman filter (EnKF) and hierarchical Bayesian inference (Zhang et al., 2024). Our main contributions are as follows:

1. We derived a full-scale single-neuron resolution weighted connectivity of the mouse brain registered to Allen Mouse Brain Common Coordinate Framework (CCFv3) (Wang et al., 2020) based on 10 $\mu$m voxel-scale axonal projections (Oh et al., 2014) and spatially resolved cell-type distributions (Zhang et al., 2023).

2. We developed the mouse DTB, a large-scale spiking neuronal network model of the mouse brain, with the inferred mouse brain connectivity on the Digital Brain platform (Lu et al., 2024a). It's validated to be able to reproduce blood-oxygen-level-dependent (BOLD) signals observed in both resting state and olfaction-based Go/No-Go discrimination task (Han et al., 2019) with a high correlation coefficient, and demonstrated correct behavioral responses consistent with perceptual odor inputs.

The mouse DTB provides a platform for in silico exploration of whole-brain dynamics and fine-scale neural mechanisms in the mouse brain.

## 2 RELATED WORKS

**Whole brain modelling.** Understanding how large-scale brain networks give rise to complex cognitive functions and behaviors has long motivated the development of whole-brain computational models (Xiong et al., 2023). By simulating whole-brain dynamics based on anatomical connectivity derived from multimodal neuroimaging data, these models provide a systematic approach for studying the mechanisms of brain function under both normal and perturbed conditions. The digital twin brain (DTB) simulates the human whole brain as a large-scale spiking neural network with up to 86 billion neurons and 47.8 trillion synapses, constructed based on multimodal structural imaging data (Lu et al., 2024a). Leveraging hierarchical mesoscale data assimilation (HMDA) method for parameter estimation, the DTB can reproduce resting-state BOLD signals with high fidelity, and accurately predicted task-evoked responses and evaluation performance (Burkitt, 2006). In parallel, the virtual brain twin (VBT) frameworks simulate individual brain dynamics with neural field models, by fit-

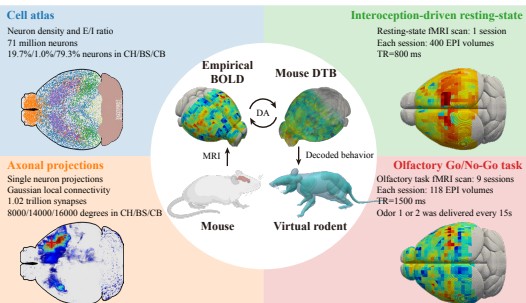

Figure 1: Overview of the mouse DTB at single-neuron resolution. The mouse DTB is based on single-neuron resolution connectivity derived from Allen Mouse Brain cell atlas and axonal projection data, and capable of reproducing BOLD signals observed in resting state and olfaction-based Go/No-Go discrimination task.

ting simulated neural activity to empirical functional recordings (Wang et al., 2024). They have been successfully applied to clinical scenarios including epilepsy and neurodegenerative diseases, where they reproduce patient-specific dynamics, predict surgical outcomes, and identify disease-relevant network alterations, supporting personalized diagnosis and treatment planning. In contrast, whole-brain modelling of the mouse remains unexplored, despite the availability of structural data including axonal projections and cell atlas (Knox et al., 2018; Oh et al., 2014). Establishing a mouse whole brain model would address this gap and enable in silico exploration of the neural mechanisms underlying cognition and behavior in the mouse brain.

**Mouse brain modelling.** Recent advances in structural imaging of the mouse brain have laid the foundation for constructing mouse brain connectivity and enabling whole brain modelling. The Allen Mouse Brain cell atlas provide cell type identities and spatial distributions of neurons across the mouse brain (Zhang et al., 2023), registered to a common coordinate framework (Wang et al., 2020). Brain-wide axonal projection data at 100 $\mu$m resolution have been mapped in the mouse brain based on systematic anterograde viral tracing experiments across hundreds of injection sites covering nearly the entire brain (Oh et al., 2014). Based on the axonal projections, a voxel-level weighted connectivity of the mouse brain at 100 $\mu$m resolution has been inferred using a kernel regression approach (Knox et al., 2018). While whole brain neuronal connectomes have been mapped in model organisms with smaller nervous systems, such as C. elegans, Drosophila, and zebrafish (Cook et al., 2019; Zheng et al., 2018; Svara et al., 2022), no single-neuron level connectivity is currently available for the mouse brain. Therefore, it's essential to construct a whole-brain connectivity at single-neuron resolution of the mouse brain, for investigating how fine-scale neural circuits support cognitive functions and behavior (Li & Wei, 2025).

**Data Assimilation.** Facing the challenge of parameter inference in large-scale neuronal network models with limited observational data, traditional Bayesian inference becomes intractable when the number of parameters far exceeds the number of observations, leading to overfitting. The hierarchical mesoscale data assimilation (HMDA) method addresses this issue by combining the diffusion ensemble Kalman filter (EnKF) with hierarchical Bayesian inference, introducing hyperparameters that govern the distribution of neuron-level parameters within each sub-unit to reduce model complexity while preserving biological structure, (Zhang et al., 2024; Lu et al., 2024a), with details in Section B.1.

## 3 METHODS

### 3.1 MODELING OF MOUSE BRAIN CONNECTIVITY AT SINGLE-NEURON RESOLUTION

We propose a data-driven pipeline for constructing a biologically plausible, weighted directed connectivity of the whole mouse brain at single-neuron resolution based on high-resolution axonal projection data (Oh et al., 2014) and spatial distribution of cells from cell atlas (Zhang et al., 2023), as depicted in Figure 2a. The pipeline comprises three main steps: (1) inference of neuronal projection connectivity based on axonal projections and spatial distribution of neurons from cell atlas, (2) iter-

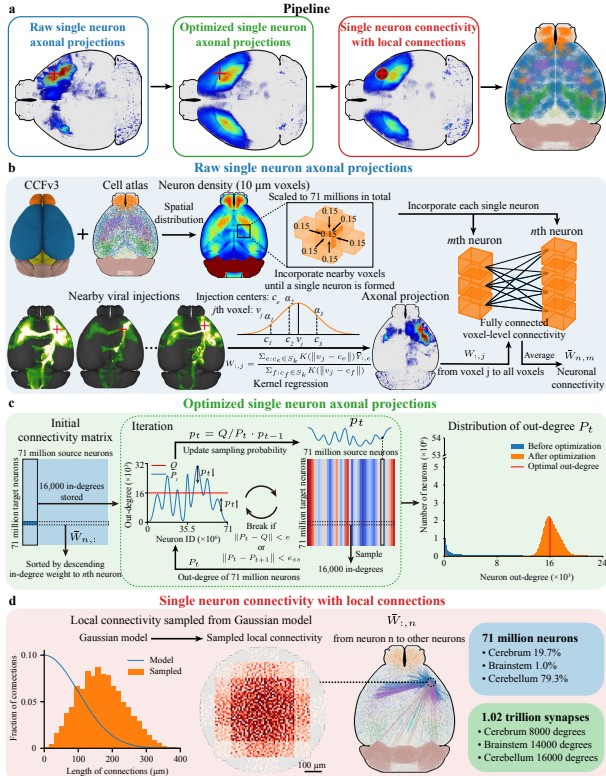

Figure 2: Modeling of mouse brain connectivity at single-neuron resolution. (a) The data-driven pipeline for constructing the mouse brain connectivity at single-neuron resolution. (b) Inference of neuronal projection connectivity. (c) Iterative optimization of neuronal projection connectivity. (d) Inclusion of Gaussian local connectivity.

ative optimization of neuronal projection connectivity to achieve topological balance in projection degrees (out-degrees) across the whole brain, and (3) inclusion of Gaussian local connectivity to compensate for missing short-range local connectivity in injection sites.

### 3.1.1 INFERENCE OF NEURONAL PROJECTION CONNECTIVITY.

**Inference of voxel-level connectivity via kernel regression.** To construct voxel-level neuronal connectivity, we utilized axonal projection data at the scale of 10 $\mu$m voxels from the Allen Mouse Brain Connectivity Atlas (Oh et al., 2014). Each of the 469 injection experiments in the dataset involves an injection vector $X$ and an projection pattern $Y$. The goal is to infer a voxel-level weighted directed adjacency matrix $W \in \mathbb{R}_{\geq 0}^{N \times N}$ such that $Y \approx WX$, where $N \approx 506$ million is the number of voxels. We made a simplified assumption that the projection weights from any given voxel vary smoothly across space, and the projections $Y$ of an injection experiment come from the center of mass $c_e$ of the injection. First, the projection density $Y_{:,e}$ from injection $e$ is normalized by the total injection density $X_{:,e}$, resulting in the normalized projection pattern $\bar{Y}_{:,e} = (Y_{:,e} + X_{:,e})/\Sigma_v X_{v,e}$. Then, as depicted in Figure 2b, we adopt kernel regression method (Knox et al., 2018) to approximate the projection weights $W_{:,j}$ from voxel $j$ to all voxels as the distance-weighted sum of the projection patterns $\bar{Y}_{:,e}$ of nearby injections:

$$W_{:,j} = \frac{\Sigma_{e:c_e \in S_k} K(\|v_j - c_e\|)\bar{Y}_{:,e}}{\Sigma_{f:c_f \in S_k} K(\|v_j - c_f\|)} \tag{1}$$

where $S_k$ is the set of centers of mass of nearby injections, $v_j$ is the position of voxel $j$, and $K(\cdot)$ adopts Gaussian radial basis function kernel.

**Reconstruction of individual neurons.** Based on the spatial distribution of neuron cells from the mouse brain cell atlas (Zhang et al., 2023), we derived the distribution of neurons across brain re-

gions, and it is further scaled according to the known regional composition of mouse brain neurons: 19.7% in the cerebrum (CH), 1.0% in the brainstem (BS), and 79.3% in the cerebellum (CB) (Erö et al., 2018). Thus, the neuron density of 10 $\mu$m voxels is assigned based on the regional neuron density, and scaled to a total of approximately 71 million neurons (Herculano-Houzel et al., 2006) across the mouse whole brain. Each voxel with incomplete neurons are incorporated with its spatially adjacent voxels using BFS, until an individual neuron is formed. As a result, individual neurons are reconstructed, which further enables the inference of neuronal connectivity from voxel-level connectivity. In addition, we derived the excitatory-to-inhibitory (E/I) ratio of neurons in each brain region based on the spatial distribution of glutamatergic and GABAergic neurons from the mouse brain cell atlas (Zhang et al., 2023). Individual neurons in each region were randomly assigned as either excitatory or inhibitory.

**Inference of neuronal connectivity.** Based on the previously inferred voxel-level connectivity and reconstruction of individual neurons, the connection weight $\bar{W}_{n,m}$ from neuron $m$ to $n$ is defined as the average of all voxel-level connection weights between the voxels occupied by neuron $m$ and $n$. As the maximum average synaptic degree of mouse brain regions is approximately 16,000 (Braitenberg & Schüz, 2013), we retain the top 16,000 in-degrees with highest weights for each individual neuron, resulting in a neuronal projection connectivity of the mouse whole brain.

### 3.1.2 ITERATIVE OPTIMIZATION OF NEURONAL PROJECTION CONNECTIVITY

Despite normalizing the projection pattern of each injection experiment by the total injection density, no normalization is performed across different experiments. As a result, brain regions exhibit highly variable synaptic out-degrees. Combined with the fact that only the top 16,000 in-degrees are retained for each neuron due to storage limitations, this leads to an extremely skewed distribution of synaptic degrees, ranging from over 47 million neurons with no outgoing connections to a small subset of neurons with more than 4 million synaptic targets. Such an imbalanced connectivity results in biologically implausible patterns of neuronal out-degrees across brain regions, and deviates significantly from experimental findings reported in the human hippocampus, where the distributions of synaptic out-degrees closely resembles that of in-degrees (Gandolfi et al., 2023). To address this inconsistency and promote a more biologically realistic and regionally balanced projection structure, we iteratively optimized the neuronal projection connectivity matrix to enforce a more uniform distribution of neuronal out-degrees across brain regions, one that approximates the distribution of in-degrees, while constraining the in-degree of each neuron to a maximum of 16,000.

As shown in Figure 2c, In iteration $t$, we compute the out-degree $P_t(n)$ of each neuron $n$ from the current connectivity matrix, and updated the sampling probability for connections originating from neuron $n$ according to $p_t(n) = Q/P_t(n) * p_{t-1}(n)$, where $Q = 16000$ is the target out-degree (equal to the fixed in-degree), and the initial value $p_0(n)$ is 1. The updated $p_t$ is capped at 1 and normalized to the sum of 1. Next, 16,000 in-degrees are sampled for each neuron based on the updated probability distribution $p_t$, yielding a new connectivity matrix with updated out-degrees $P_{t+1}$. The iterative optimization process continues until convergence, defined as either $\|P_t - Q\| < e$ or $\|P_t - P_{t+1}\| < e_{ss}$, where $e$ is the target error and $e_s s$ is the steady state error. Through the iterative optimization procedure, the sampling probability of neurons with excessive out-degrees is progressively reduced, allowing their connections to be redistributed toward neurons with insufficient out-degrees. This results in a more balanced projection structure across brain regions and yields a distribution of synaptic out-degrees that closely resemble that of in-degrees. For the optimized connectivity, the top 8,000, 14,000, and 16,000 in-degree connections with the highest weights are retained for neurons in CH, BS and CB, respectively (Braitenberg & Schüz, 2013).

### 3.1.3 GAUSSIAN LOCAL CONNECTIVITY

Due to the relatively large injection volume ($\sim$0.24 mm$^3$ on average), short-range axonal projections within the injection site are often obscured and thus underrepresented in the observed projection data. To compensate for this loss and restore the intrinsic local connectivity, we introduce Gaussian-distributed local connections among spatially proximal neurons (Potjans & Diesmann, 2014; Schmidt et al., 2018; Campagnola et al., 2022), as depicted in Figure 2d. These connections are sampled according to Gaussian local connectivity model, $C(r) = C_0 exp(\frac{-r^2}{2\sigma^2})$ (Potjans & Diesmann, 2014), where $r$ denotes the distance between neurons. The parameters are set to $C_0 = 0.1$ and $\sigma = 100\mu$m (Campagnola et al., 2022), with sampling constrained within a 300 $\mu$m radius.

These sampled connections compensate for the loss of local projections and contribute to a more complete and biologically faithful reconstruction of the mouse brain connectivity.

### 3.2 SIMULATION AND ASSIMILATION OF MOUSE DTB

**Simulation.** We simulate a large-scale spiking neural network of the mouse DTB on the DB platform (Lu et al., 2024a). Individual neurons are modeled based on leaky integrate-and-fire (LIF) neuron model and connected via conductance-based synapses with exponential postsynaptic currents (Burkitt, 2006), which includes 4 types of synapses: AMPA and NMDA for excitatory synapses, and GABA$_A$ and GABA$_B$ for inhibitory synapses. Simulations of the full-scale mouse DTB with approximately 71 million neurons and 1.02 trillion synapses are performed on a GPU-based high-performance computing (HPC) system comprising 160 nodes and 640 GPUs. This simulation framework enables efficient large-scale simulation and assimilation experiments for the mouse DTB.

**Assimilation.** We adopt the HMDA method (Zhang et al., 2024) to infer voxel-wise synaptic parameters of the mouse DTB from observed BOLD signals. In resting-state and task experiments, we assume that neural activity in task-relevant perceptual regions of interest (ROIs) drives whole-brain dynamics. Therefore, we infer the input currents of neurons in the perceptive ROIs by assimilating voxel-level BOLD signals of ROIs. These assimilated currents are then injected into the mouse DTB model, enabling stimulus-evoked neural activity propagation across the entire brain.

## 4 RESULTS

### 4.1 STRUCTURE OF MOUSE BRAIN CONNECTIVITY AT SINGLE-NEURON RESOLUTION

**Spatial distribution of neurons.** The average neuronal densities across 580 mouse brain regions range from 71.8 neurons/mm$^3$ ( in a region of the pons, $P$), to 2247765.9 neurons/mm$^3$ (in a region of CB), as shown in Figure 3(a). The mean neuronal densities of brain regions in CH, BS and CB are 41759.3±19746.2, 6439.4±7102.8 and 804863.3±788219.1 neurons/mm$^3$, respectively, revealing substantial variability in spatial distribution of neurons at the whole-brain scale. Across the whole brain, excitatory neurons account for 60.68% and inhibitory neurons for 39.32% of the total neuronal population. The fraction of excitatory neurons varies substantially across brain regions, ranging from 0 to 1.

**Connectivity.** The initial neuronal projection connectivity of the mouse brain exhibits substantial heterogeneity, with average regional out-degrees ranging from 3.8 to 1,471,999.7 and a global mean of 33,844.4, as depicted in Figure 3(b). This wide range of projection degree reflects a highly unbalanced projection structure, where a small number of regions dominate outgoing connections. To address this imbalance, we applied the iterative optimization procedure to balance the distribution of out-degree across regions. As a result, the average out-degree of region was adjusted to 15,833.0±263.0, closely aligning with the fixed in-degree and achieving a balanced projection pattern across the whole brain. In the resulting mouse brain connectivity, inter-regional connection strengths span approximately 10 orders of magnitude, encompassing both sparse long-range projections and dense local subnetworks, as shown in Figure 3(c). Notably, strong bidirectional connectivity is observed within the cerebrum, within the cerebellum, and between these two major brain regions. In contrast, the initial neuronal projection connectivity is dominated by a small set of regions with disproportionately high out-degrees, leading to a highly imbalanced structure that disregards many biologically important connections and pathways. This suggest that the optimized connectivity successfully captures biologically plausible patterns of inter-regional communication by promoting global balance in projection degrees.

**Activity propagation in mouse DTB.** To experimentally probe the structural organization of neural circuits in the mouse DTB, we applied stimulus current into specific brain regions and analyzed the spatiotemporal propagation of neural activity. To quantify and visualize the propagation of neural activity, we the total energy of the response variations in the average firing rate of each voxel, defined as $E = \Sigma_t |r(t)|^2$, where r(t) denotes the average firing rate at time-step $t$ (Qi et al., 2024). Stimulation was delivered to neurons within a target brain region using a 0.2 nA input current lasting 800 ms, after which the stimuli were removed. We then computed the time series of energy $E(t)$, using a sliding window of 100 ms with a step size of 1 ms. Through this approach, we are able to

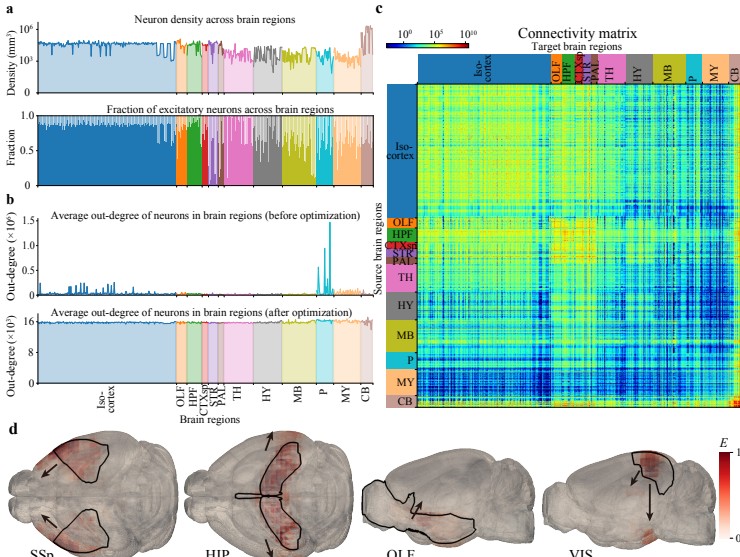

Figure 3: Structure of the mouse brain connectivity at single-neuron resolution. (a) Spatial distribution of neurons. (b) Distribution of projection degrees before and after iterative optimization. (c) Connectivity between brain regions. (d) Neural activity propagation across the mouse DTB.

track the propagation of neural responses across the whole brain following stimulus removal, and identify the spatial structure of underlying neural pathways, as shown in Figure 3(d). The observed activity patterns reflect both direct anatomical projections and indirect multi-synaptic pathways that are difficult to infer directly from structural connectivity, with detailed analysis in Section D.4.

## 4.2 MOUSE DTB IN RESTING STATE

**Rest-state fMRI data.** The empirical resting-state fMRI data consisted of 400 contiguous EPI volumes with a spatial resolution of $400\mu m \times 400 \ \mu m \times 400 \ \mu m$, covering 6,131 voxels in the mouse brain. To enable voxel-wise comparison of BOLD signals, we grouped the neurons in the mosue DTB model into neuron populations with the same spatial resolution, resulting in 8,930 voxel populations. Neural activity within each population was aggregated to generate simulated BOLD signals at the same resolution as the empirical data.

**Interoception-driven resting state.** The afferent interoceptive pathways are known to convey information about the internal physiological state of the body, and are thought to play a critical role in shaping intrinsic brain activity (Berntson et al., 2019), as shown in Figure 4(a). To investigate the influence of interoceptive circuits on the resting state, we simulated the spontaneous brain dynamics in the mouse DTB under interoception-driven input. Specifically, we assimilated the input currents of neurons in key interoceptive regions, including SSp, HIP, Amy, AI, mPFC, TH, HY and PAG (Berntson & Khalsa, 2021), by fitting their empirical BOLD signals, as depicted in Figure 4(b). We achieved a high average correlation coefficient of 0.948 across 2,147 voxels within these regions, indicating accurate estimation of interoception-driven inputs. Driven by the assimilated currents, the mouse DTB successfully reproduced resting-state BOLD signals across the whole brain, reaching an average correlation coefficient of 0.901 with empirical BOLD signals across all 6,131 voxels, as depicted in Figure 4(c). These results demonstrate that the interoception-driven mouse DTB can effectively replicate the spatiotemporal patterns of spontaneous whole-brain neural activity observed in the mouse resting state.

**Effect of synaptic degree on interoception-driven resting state.** We conducted interoception-driven resting-state simulations on mouse DTB models with varying synaptic degrees of 200, 2,000, 4,000, 8,000, 12,000, and 16,000. As the synaptic degree increased, the average correlation coefficient between simulated and empirical BOLD signals across the whole brain improved steadily, rising from 0.835 at a degree of 200 to 0.901 at 16,000, as shown in Figure 4(d). This indicates that

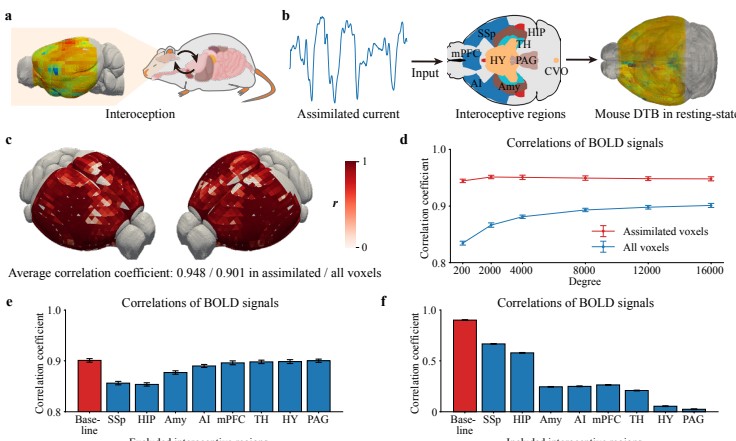

Figure 4: Mouse DTB in resting state. (a) Resting-state of mouse. (b) Framework of interoception-driven resting-state experiments. (c) Average correlation coefficient of the mouse DTB in resting-state. (d) Effect of synaptic degree on interoception-driven resting state. (e) Effect of excluded interoceptive regions on interoception-driven resting state. (f) Effect of individual interoceptive region on interoception-driven resting state.

more biologically realistic connectivity enables more effective propagation of interoceptive inputs and better supports the emergence of large-scale resting-state dynamics in the mouse DTB.

**Effect of interoceptive regions on interoception-driven resting state.** To evaluate the relative importance of individual interoceptive regions in driving resting-state brain dynamics, we conducted two sets of ablation experiments: (1) removing the input to one interoceptive region while keeping the others active, and (2) providing input to only one interoceptive region at a time. SSp and HIP emerged as the most critical interoceptive regions, as illustrated in Figure 4(e-f). Removing either region caused the largest reduction in whole-brain performance, while driving the DTB with only one of them resulted in the strongest individual effects. These results indicate that both regions play essential roles in mediating interoception-driven resting-state dynamics. Conversely, HY and PAG showed minimal impact on resting-state dynamics, regardless of whether they were removed or served as the sole source of input.

### 4.3 MOUSE DTB IN ACTION

**Olfactory Go/No-Go discrimination task.** Olfactory discrimination is a well-established decision-making task, involving in sensory processing, reward expectation, motor planning, and inhibitory control (Han et al., 2019). It serves as a model paradigm for examining how the brain integrates sensory cues and behavioral rules to generate goal-directed actions under temporally structured conditions.

In the olfactory Go/No-Go discrimination task, mice were trained to distinguish between two odors (odor 1 and 2) and respond with a corresponding lick or no-lick behavior, as shown in Figure 5(a). During scanning, odor stimuli were delivered every ∼15 s for the duration of 1 s, and both fMRI data and behavioral responses were recorded (Han et al., 2019). A total of 9 sessions from a single subject were collected from a single subject, each consisted of 118 contiguous EPI volumes with the spatial resolution, covering 6,800 voxels in the mouse brain.

**Olfactory task state.** To investigate the neural mechanisms underlying the olfactory Go/No-Go discrimination task, we simulated olfactory task-state brain dynamics in the mouse DTB. Specifically, we assumed that the task-evoked dynamics are primarily driven by olfactory sensory input and partially contributed by from interoceptive input. Therefore, we assimilated the input currents of neurons in OLF and two significant interoceptive regions, SSp and HIP, identified in previous experiments, as depicted in Figure 5(b). Assimilation was performed by fitting the empirical BOLD signals within these regions, achieving a high average correlation coefficient of 0.948 across 2,002 voxels.

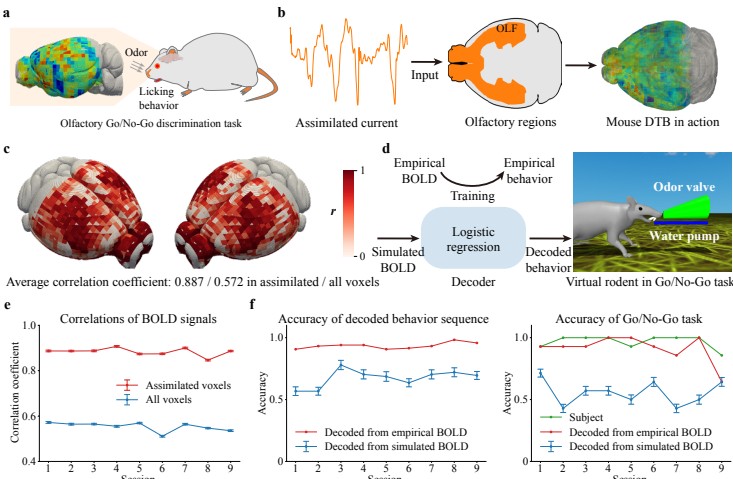

Figure 5: Mouse DTB in action. (a) Olfactory Go/No-Go discrimination task. (b) Framework of olfactory task state experiments. (c) Average correlation coefficient of the mouse DTB in action. (d) Go/No-Go behavior decoding. (e) Correlation coefficients across all sessions. (f) Accuracy of behavior decoded from simulated BOLD generated by mouse DTB.

Driven by the assimilated olfactory sensory and interoceptive inputs, the mouse DTB successfully reproduced task-state BOLD signals across the whole brain, achieving an average correlation coefficient of 0.901 across 6,800 voxels in session 1, as shown in Figure 5(c). Across all 9 sessions, the mouse DTB achieved an average whole-brain correlation of 0.554±0.019, as shown in Figure 5(e). These results indicate that the mouse DTB can robustly capture the spatiotemporal dynamics of brain-wide activity during olfactory decision-making.

**Go/No-Go behavior decoding.** To evaluate whether the simulated task-state dynamics in the mouse DTB can support behavior prediction, we trained a logistic regression decoder to classify binary Go/No-Go behavioral responses of the mouse DTB. The decoder was trained using the empirical task-state fMRI data and the corresponding behavioral sequences, and was then applied to decode the simulated BOLD signals generated by the mouse DTB, yielding a predicted sequence of behavioral responses, as shown in Figure 5(d). The predicted sequence of Go/No-Go behavior is further simulated in the virtual rodent environment (Merel et al., 2019; Aldarondo et al., 2024).

To validate the fidelity of the behavior generated by the mouse DTB, we compared the decoded behavioral sequence with the empirical behavioral responses. Two metrics were used for evaluation: the accuracy of overall sequence, and the Go/No-Go discrimination accuracy under odor stimulus onset. Across 9 sessions, the DTB achieved an average sequence accuracy of 67.33±6.64%, and an average odor discrimination accuracy of 55.56±9.39%, as shown in Figure 5(f). While the decoding performance does not yet reach the level of empirical prediction models, the above-chance accuracy suggests that the simulated dynamics capture task-relevant neural representations to a meaningful extent. The suboptimal performance may be attributed to insufficient assimilation accuracy in brain regions critical for olfactory-guided behavior, limiting the decoder's ability to recover fine-grained behavioral patterns from the simulated dynamics.

## 5 CONCLUSION

In this study, we deliver a biologically realistic, brain-wide, neuronal-level mouse DTB model that bridges structural connectivity and dynamic function across the whole brain. By validating the model's ability to reproduce BOLD signals and intelligent behavioral responses observed in resting-state and in action, we highlight its potential as a general platform for in silico exploration of neural dynamics and emergent functions of the mouse brain. Our framework also provides new opportunities to investigate fine-grained neural mechanisms of individual neurons across the whole brain with experimentally recorded neuronal activity (de Vries et al., 2020; Xue et al., 2024; Lai et al., 2023).

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

## A   SIMULATION OF MOUSE DTB

We performed large-scale spiking neural network simulations of the mouse digital twin brain (DTB) on the Digital Brain (DB) platform (Lu et al., 2024a;b). The mouse DTB is constructed based on the inferred mouse brain connectivity at single-neuron resolution, and modeled using leaky integrate-and-fire (LIF) neurons and conductance-based synapses (Burkitt, 2006).

## A.1 HARDWARE AND SOFTWARE ENVIRONMENT

All mouse DTB simulations and assimilations were executed on a Linux-based high-performance computing (HPC) cluster, comprising 160 compute nodes and 640 GPUs. Each compute node is equipped with a single 32-core CPU operating at 2.0 GHz and 128 GB of DRAM. Additionally, each node includes 4 GPUs operating at 1.10 GHz, each with 16 GB of HBM2 memory running at 800 MHz, and delivering up to 1 TB/s of memory bandwidth. GPUs within the same node communicate via shared memory, while inter-node GPU communication is performed over a 200 Gbps full-duplex InfiniBand network. The C++ simulation framework was compiled using `g++` version 7.3.1 with support for MPI-based distributed communication (Barker, 2015) and GPU acceleration via `hipcc` from the ROCm 4.1.0 toolkit. Simulation and assimilation of the mouse DTB were implemented in Python (version 3.7.11).

## A.2 COMPUTATIONAL MODEL OF THE MOUSE DTB

The mouse DTB contains approximately 71 million neurons and 1.02 trillion synapses, constructed based on the inferred mouse brain connectivity at single-neuron resolution. Each neuron in the mouse DTB follows a leaky integrate-and-fire (LIF) model with conductance-based synaptic inputs (Burkitt, 2006). The membrane potential $V_i$ of neuron $i$ evolves according to:

$$C_i \frac{dV_i}{dt} = -g_{L,i}(V_i - V_L) + \sum_u I_{\text{syn},i} + I_{\text{bg},i} + I_{\text{ext},i}, \quad V_i < V_{\text{th},i} \tag{2}$$

where:

- $C_i$ is the neuron membrane capacitance,
- $g_{L,i}$ is the leakage conductance,
- $V_L$ is the leakage voltage,
- $I_{\text{syn},i}$ is the total synaptic current from four types of synapses (AMPA, NMDA, GABA$_A$, GABA$_B$),
- $I_{\text{bg},i}$ is the background current serving as noises, and
- $I_{\text{ext},i}$ is the external current serving as the external injection stimuli for tasks, which are independently sampled from a Gamma distribution with the assimilated hyperparameters.

Each spike causes a discontinuous reset. When $V_i = V_{\text{th},i}$ at time $t_n^i$, a spike is emitted, and the potential is reset to $V_{\text{rest}}$ for a refractory period.

In the mouse DTB, four synapse types are considered, which are AMPA, NMDA, GABA$_A$, and GABA$_B$. The synaptic input current of type $u$ from neuron $j$ to neuron $i$, is described as:

$$I_{u,i} = g_{u,i}(V_u - V_i)J_{u,i} \tag{3}$$

$$\frac{dJ_{u,i}}{dt} = -\frac{J_{u,i}}{\tau_{u,i}} + \sum_{k,j} w_{i,j}^u \delta(t - t_k^j) \tag{4}$$

Here, $g_{u,i}$ is the maximal synaptic conductance, $V_u$ is the voltage, $\tau_{u,i}$ is the decay constant, $w_{i,j}^u$ is the connection weight from neuron $j$ to neuron $i$, $\delta(\cdot)$ is the Dirac-delta function, and $t_k^j$ are presynaptic spike times.

The background current is modeled by an Ornstein–Uhlenbeck (OU) processes:

$$\tau_{\text{bg}} dI_{\text{bg},i} = (\mu_{\text{bg}} - I_{\text{bg},i})dt + \sqrt{2\tau_{\text{bg}}} \, \sigma_{\text{bg}} dW_{i,t} \tag{5}$$

where $\tau_{\text{bg}}$, $\mu_{\text{bg}}$, and $\sigma_{\text{bg}}$ are time constant, mean, and standard deviation shared across neurons, and $W_{i,t}$ is standard Brownian motion.

The parameters used in the computational neuron and synapse model and their corresponding values are listed in Table1. All numerical updates of neuron states are based on a first-order Euler-Maruyama method with a time-step of 1 ms. Spike communication across GPUs is achieved with message-passing interface (MPI) (Barker, 2015), and synchronized at 1 ms intervals.

Table 1: Parameters used in the computational neuron and synapse model.

| Category | Parameter | Symbol | Value |
|---|---|---|---|
| Neuron model | Membrane capacitance | $C$ | 0.5 nF |
| | Leak conductance | $g_L$ | 25 nS |
| | Leak reversal potential | $V_L$ | $-70$ mV |
| | Spiking threshold | $V_{\text{threshold}}$ | $-50$ mV |
| | Reset potential | $V_{\text{reset}}$ | $-55$ mV |
| | Refractory period | $T_{\text{ref}}$ | 2 ms |
| Synapse model | AMPA reversal potential | $V_{\text{AMPA}}$ | 0 mV |
| | GABA reversal potential | $V_{\text{GABA}}$ | $-70$ mV |
| | AMPA time constant | $\tau_{\text{AMPA}}$ | 2 ms |
| | GABA time constant | $\tau_{\text{GABA}}$ | 20 ms |
| | AMPA synaptic conductance | $g_{\text{AMPA}}$ | 2 nS |
| | GABA synaptic conductance | $g_{\text{GABA}}$ | 10 nS |
| | Range of synaptic weight | $w_{ij}$ | $[0, 1]$ |
| OU background current | Mean background current | $\mu_{\text{bg}}$ | 0.4 nA |
| | Std. dev. of background current | $\sigma_{\text{bg}}$ | 0.15 nA |
| | Background current time constant | $\tau_{\text{bg}}$ | 4 ms |

### A.3 BALLOON–WINDKESSEL MODEL

We employed the Balloon-Windkessel hemodynamic model to generate simulated BOLD signals from the neural activities of the mouse DTB (Friston et al., 2000). This model provides a biophysically grounded mapping from neural activity to the blood-oxygen-level-dependent (BOLD) response by modeling neurovascular coupling dynamics at the voxel level. Specifically, the population-averaged firing rate of each voxel, denoted by $z_i(t)$ for voxel $i$, is transformed into the corresponding BOLD signal $y_i(t)$. The dynamics of this model is governed by the following system of coupled differential equations:

$$\dot{s}_i = z_i - \kappa_i s_i - \gamma_i(f_i - 1) \tag{6}$$

$$\dot{f}_i = s_i \tag{7}$$

$$\tau_i \dot{v}_i = f_i - v_i^{\frac{1}{\alpha}} \tag{8}$$

$$\tau_i \dot{q}_i = \frac{f_i E(f_i, \rho_i)}{\rho_i} - v_i^{1/\alpha} q_i / v_i \tag{9}$$

$$y_i = V_0 \left[ k_1(1 - q_i) + k_2(1 - \frac{q_i}{v_i}) + k_3(1 - v_i) \right] \tag{10}$$

where:

- $s_i$ is the vasodilatory signal of voxel $i$,
- $z_i$ is the neural activity (firing rate) of voxel $i$,
- $f_i$ is the blood inflow,
- $v_i$ is the blood volume,
- $q_i$ is the deoxyhemoglobin content,
- $E(f_i, \rho_i) = 1 - (1 - \rho_i)^{1/f_i}$ is the fraction of oxygen extracted from the inflowing blood,
- $y_i$ is the BOLD signal.

The model parameters are set to standard physiological value:

- $\kappa_i = 1.25$ is the the inverse of the decay time constant of the vasodilatory signal $s_i$,
- $\gamma_i = 2.5$ is the the inverse of the time constant of the inflow $f_i$,

Table 2: Simulation performance of the mouse DTB with different synaptic degrees in resting-state.

| Nodes | GPUs | CH degree | BS degree | CB degree | Firing rate (Hz) | Real-time factor |
|-------|------|-----------|-----------|-----------|------------------|------------------|
| 2 | 8 | 100 | 175 | 200 | 4.237 | 19.90 |
| 20 | 80 | 1,000 | 1,750 | 2,000 | 4.261 | 8.56 |
| 40 | 160 | 2,000 | 3,500 | 4,000 | 4.220 | 8.76 |
| 80 | 320 | 4,000 | 7,000 | 8,000 | 4.201 | 10.18 |
| 120 | 480 | 6,000 | 10,500 | 12,000 | 4.207 | 11.19 |
| 160 | 640 | 8,000 | 14,000 | 16,000 | 4.251 | 12.96 |

- $\tau_i = 1$ is the the time constant of the blood volume $v_i$, which is the same with that of the deoxyhemoglobin content $q_i$,
- $\alpha = 0.2$ is a stiffness exponent which specifies the flow–volume relationship of the venous balloon,
- $\rho_i = 0.8$ is the the resting oxygen extraction fraction
- $V_0 = 0.02$ is the the resting blood volume fraction, and
- $k_1 = 7\rho_i$, $k_2 = 2$, $k_3 = 2\rho_i - 0.2$ are the BOLD signal coefficients.

Simulated BOLD time series were generated for both resting-state and task-state conditions, and compared to empirical fMRI data using voxel-wise Pearson correlation. These signals were also used as input for data assimilation and behavioral decoding, providing a consistent observation model across the entire mouse DTB simulation pipeline.

### A.4 SIMULATION PERFORMANCE OF THE MOUSE DTB

In order to evaluate the effect of synaptic degree on interoception-driven resting state, we simulated the mouse DTB in resting-state, under varying synaptic degrees, as shown in Figure 4d and discussed in Section 4.2. The detailed configuration of synaptic degrees and the computational efficiency of the mouse DTB is displayed in Table 2. For simulations with average firing rates of approximately 4.2 Hz, we have achieved a performance in which 1 s of biological time requires 19.90 s and 12.96 s of computation time, corresponding with real-time factors of 19.90 and 12.96, respectively. This demonstrates that even at full biological resolution, the mouse DTB can be simulated efficiently in near real-time, enabling scalable exploration of neural dynamics of the mouse whole brain.

## B ASSIMILATION OF MOUSE DTB

We employed the Hierarchical Mesoscale Data Assimilation (HMDA) method, which integrates a diffusion Ensemble Kalman Filter (EnKF) with hierarchical Bayesian inference, to enable parameter estimation in large-scale neuronal network models under conditions of limited observational data (Zhang et al., 2024; Lu et al., 2024a).

### B.1 FRAMEWORK OF THE HMDA METHOD

In high-dimensional neuronal network models, the number of parameters to be inferred often far exceeds the number of available data points. This imbalance renders canonical Bayesian inference intractable and prone to overfitting. To overcome this challenge, we adopt a hierarchical Bayesian inference framework by introducing hyperparameters that define the prior distribution of model parameters, as illustrated in Figure 6.

Let $y_t$ denote the observed data (e.g., BOLD signals), $x_t$ the internal latent states (e.g., membrane potentials, synaptic currents, and hemodynamic variables), $\theta$ the model parameters (e.g., synaptic conductances and external currents), and $\vartheta$ the hyperparameters that govern the distribution of $\theta$. The joint posterior of the model can be expressed as:

$$P(\vartheta \mid y_t) \propto \int P(y_t \mid x_t, \theta)\, P(x_t \mid \theta)\, P(\theta \mid \vartheta)\, P(\vartheta)\, dx_t\, d\theta \tag{11}$$

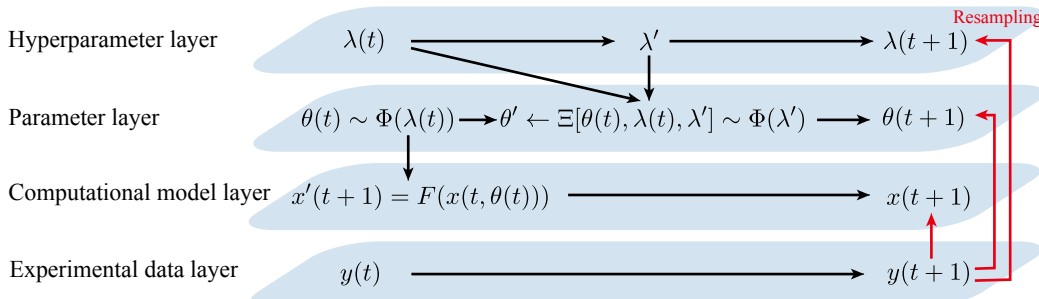

Figure 6: Framework of the hierarchical Bayesian inference.

This hierarchical formulation introduces an intermediate layer between parameters and their priors, allowing parameter values to be flexibly adapted at the voxel level while being regularized by voxel-level hyperparameters. By learning $\vartheta$ instead of directly estimating every element of $\theta$, the number of free variables is significantly reduced, thereby alleviating overfitting while preserving biological heterogeneity across the brain. The balance between model expressiveness and statistical tractability is achieved by simultaneously inferring both the hyperparameters and the parametric distribution of the parameters.

In addition, we considered neurons within each $400\times400\times400$ $\mu m^3$ voxel as an individual neuron population, and assumed that neurons of the same type (e.g., excitatory and inhibitory) within a population shared the same hyperparameters. The time-series BOLD signals of the voxels were used as observations of the mouse DTB.

The framework consists of four layers. The **hyperparameter layer** performs a random walk update of the region-level hyperparameters from $\lambda(t)$ to $\lambda'$. The **parameter layer** samples the parameter vector $\theta(t)$ from $\lambda(t)$ using a sampling operator $\Phi(\cdot)$, and modifies it based on the change from $\lambda(t)$ to $\lambda'$, yielding $\theta(t+1)$. The **computational model layer** evolves the hidden neural state $x(t)$ by integrating the dynamical system $\dot{x} = F(x, \theta)$ parameterized by $\theta(t)$. The **experimental data layer** generates the observation $y(t+1)$ from the updated hidden state $x(t+1)$ and uses it to update both the hidden state and the parameters, and importantly, to resample the hyperparameters from $\lambda'$ to $\lambda(t+1)$.

The computational complexity of each EnKF update scales with the dimension of the observations. What's more, combining high-resolution experimental BOLD signals with a limited number of time points results in both high computational cost and potential ill-posedness. To address this, we adopted diffusion ensemble Kalman filter, treated each voxel-level BOLD signal as an independent observer and established an independent EnKF, as demonstrated in Algorithm 1. The correction of states, parameters, and hyperparameters for each voxel is computed as a weighted average of its own EnKF update and those of other voxels. This diffusion-based update is controlled by a fusion coefficient $\gamma$ for balancing the fusion of corrections from itself and all others.

### B.2 EVALUATION METRICS

To quantitatively evaluate the fidelity of simulated BOLD signals produced by the mouse DTB, we employed voxel-wise Pearson correlation coefficients (PCC) as the primary metric. Specifically, for each voxel, we calculated the correlation between the time series of simulated BOLD signals and the corresponding empirical BOLD signals. This voxel-level PCC reflects how well the temporal dynamics of the digital model align with empirical brain activity.

Due to the inherent latency introduced by the data assimilation process, there is often a slight delay in tracking fast-varying signals. To compensate for this, we adopt a lagged version of the PCC, defined as:

$$\text{PCC}(y_{\text{DTB}}(t + \text{lag}), \ y_{\text{emp}}(t)) \tag{12}$$

where $y_{\text{DTB}}(t)$ and $y_{\text{emp}}(t)$ denote the simulated and experimental BOLD signals, respectively, and lag= 3 time steps.

---

**Algorithm 1** Diffusion Ensemble Kalman Filter (EnKF) with hierarchical Bayesian inference.

1: **Input:**
2: Ensemble member number $N$,
3: Initial distribution parameters $[\mu_0, C_0]$,
4: Evolution function $F_{0:T}(\cdot)$,
5: Hyperparameter random walk covariance matrix $\Sigma_h$,
6: Model noise covariance matrix $\Sigma_x$,
7: Parameters evolution noise $\eta_{0:T}$,
8: Observation matrix $H$,
9: Observation error covariance matrix $\Gamma$,
10: Observation (voxel-wise BOLD signals in ROIs) $y_{0:T}$,
11: Fusion coefficient $\gamma$.
12: **Output:** State estimation $\{X_0^n : T\}_{n=1}^N$
13: **draw** $X_0^{(n)} \sim \mathcal{N}(\mu_0, C_0)$, $\Theta_0^{(n)} \sim \psi(\cdot | \bar{h}_0^{(n)})$, $\hat{h}_0^{(n)} = h_0^{(n)}$, $\forall n = 1 : N$
14: **for** $t = 1 : T$ **do**
15:     **draw** $\hat{h}_t^{(n)} \sim \mathcal{N}(h_{t-1}^{(n)}, \Sigma_h)$, $\forall n = 1 : N$
16:     $\Theta_t^{(n)} = \Phi(\Theta_{t-1}^{(n)}, \hat{h}_t^{(n)}, \hat{h}_{t-1}^{(n)}, \eta_t)$, $\forall n = 1 : N$
17:     **draw** $\hat{X}_t^{(n)} \sim \mathcal{N}(F_{t-1}(X_{t-1}^{(n)}, \Theta_t^{(n)}), \Sigma_x)$, $\forall n = 1 : N$
18:     **compute** $\bar{X}_t = \frac{1}{N} \sum_{n=1}^N \hat{X}_t^{(n)}$
19:     $\hat{C}_t = \frac{1}{N-1} \sum_{n=1}^N (\hat{X}_t^{(n)} - \bar{X}_t)(\hat{X}_t^{(n)} - \bar{X}_t)^\top$
20:     **derive Kalman gain** $S_t = H\hat{C}_t H^\top + \Gamma$
21:     $K_t = \hat{C}_t H^\top S_t^{-1}$
22:     **draw** $\Delta y_t^{(n)} \sim \mathcal{N}(0, \Gamma)$, $\forall n = 1 : N$
23:     $\delta_t^{(n)} = y_t + \Delta y_t^{(n)} - H\hat{X}_t^{(n)}$, $\forall n = 1 : N$
24:     **filter by** $X_{t+1/2}^{(n)} = \hat{X}_t^{(n)} + K_t \delta_t^{(n)}$, $\forall n = 1 : N$
25:     **correct** $X_t^{(n)} = diag(X_{t+1/2}^{(n)} \Upsilon(\gamma))$, **where** $[\Upsilon(\gamma)]_{i,i} = \gamma$, **and** $[\Upsilon(\gamma)]_{i,j} = \frac{1-\gamma}{L-1}$, $i \neq j$
    **return** $\{X_0^n : T\}_{n=1}^N$

---

### B.3 Assimilation of mouse DTB in resting-state

As the HMDA framework allows joint estimation of all hidden states and parameters in the model, we made the simplifying assumption that the network dynamics are primarily driven by variations in maximum synaptic conductance under resting-state condition. Accordingly, we assimilate experimental resting-state BOLD signals to estimate the voxel-wise hyperparameters governing synaptic conductance $g_{u,i}$ of each voxel $i$, where $u = \text{AMPA}$ is the assimilated synapse type. After assimilation of the hyperparameters, we resimulated the mouse DTB by assigning the maximum synaptic conductance to each neuron based on samples drawn from the inferred hyperparameter distributions.

Specifically, assimilation with ensemble number $N = 50$ is conducted on mouse DTB with 100, 175 and 200 synaptic degrees in CH, BS and CB, respectively. Next, we conducted simulations of the mouse DTB with different synaptic degrees, by employing the assimilated hyperparameters. For a given average synaptic degree $D$, the synaptic conductance of type $u$ assigned to neuron $i$ is scaled from the original estimated value at degree $d$ as:

$$g_{u,i,D} = g_{u,i,d} \cdot \frac{D}{d} \tag{13}$$

Resting-state BOLD signals were assimilated using this model configuration to estimate synaptic conductance parameters of the mouse DTB. The resulting mouse DTB in resting-state is used for stimulation experiments demonstrated in Section 4.1 and Figure 3d. Additionally, the deployed mouse DTB comprises 100, 175 and 200 synaptic degrees in CH, BS and CB, respectively, and the computed energy of each stimulation experiment are based on firing rates averaged from 300 repeated simulations.

### B.4 Assimilation of mouse DTB in interoception-driven resting-state and in action

For assimilations of mouse DTB in interoception-driven resting-state and in action, we adopted a dual inference strategy: (1) the hyperparameters of maximum synaptic conductance $g_{u,i}$ in the mouse whole brain were estimated from resting-state fMRI data, as discussed in previous section, (2) the hyperparameters of the external currents $I_{\text{ext},i}$ injected into voxels within ROIs were inferred from the corresponding resting-state and task-state fMRI data. Specifically, we focused on estimating the hyperparameters of $I_{\text{ext},i}$ for neurons in voxels within the perceptive ROIs. In the case for interoception-driven resting-state, brain regions SSp, HIP, Amy, AI, mPFC, TH, HY and PAG are considered, while for olfactory task-state, additional brain region OLF is considered.

We applied the HMDA method on the voxels within corresponding perceptive ROIs to decode the sensory stimulus. The assimilated hyperparameters of the voxels within perceptive ROIs were then used for sampling external currents injected into perceptive ROIs during re-simulation, which drove stimuli-evoked neural activity in the mouse DTB. For each session of resting-state and task-state fMRI data, we conducted 10 repeated simulations and computed the averaged voxel-wise PCC.

## C Experiments on mouse DTB

### C.1 Procedure of mouse DTB experiments

Detail description of our mouse DTB model's experiment procedure are as follows:

1. **Data assimilation ("training")**: We estimated the parameters governing the stimulation currents into the olfactory and key interoceptive regions, by fitting simulated BOLD signals to empirical BOLD signals in these regions using the HMDA method, described in detail in Supplementary Section B. As a result, driven by the estimated stimulation currents, the simulated and empirical BOLD signals in the stimulated regions exhibit high correlation, and we can assume that the stimulation input of olfactory signals and interoceptive signals is faithfully reproduced in our computational model.

2. **Simulation (testing)**: Stimulated by the estimated currents, the stimulated regions (olfactory and key interoceptive regions) will then drive whole-brain neural activity in the mouse

Table 3: Comparison between traditional machine learning models and the Mouse DTB model.

|  | Traditional ML Model | Mouse DTB Model |
|---|---|---|
| Training Input | Training data with input-output pairs | Empirical BOLD signals in stimulated brain regions |
| Training Target | Model weights | Parameters governing the stimulation currents into the olfactory and key interoceptive regions |
| Training Goal | Fit output labels | Fit simulated BOLD to empirical BOLD in stimulated regions |
| Training Method | Supervised learning (loss minimization) | Data assimilation (the HMDA method) |
| Testing | Test on hold-out input-output pairs | Predict whole-brain BOLD responses outside the stimulated regions driven by the estimated stimulation currents |
| Evaluation | Prediction accuracy on test set | Correlation between simulated (predicted) and empirical BOLD in non-stimulated regions |
| Assumption | Train/test are i.i.d. or split by condition | Stimulated regions drive downstream activity across the whole brain |

DTB model, and then we evaluate how well the simulated (predicted) BOLD responses in the rest of the brain match the empirical BOLD signals.

3. **The goal of assimilation and simulation**: Through the process of assimilation and simulation, we can verify if the mouse DTB can reproduce the correct brain-wide BOLD responses when driven by the estimated stimulation currents. Therefore, we can validate the biological plausibility of the mouse brain connectivity at single-neuron, and the effectiveness of our mouse DTB model.

4. **Behavior decoding**: We would further decode the olfactory discrimination behavior based on the simulated BOLD signals of mouse DTB model to ascertain whether the neural activity of our model successfully encodes olfactory behavior. This allows us to assess the model's capability to generate the correct whole brain dynamics under localized stimulation.

### C.2 METHODOLOGY OF MOUSE DTB EXPERIMENTS

In conventional machine learning methodology, neural network models are trained on train sets to match the output, and later tested on hold-out test sets for their performance in predicting the correct output. In contrast, in our mouse DTB model, parameters are fitted to match the BOLD signals in the stimulated regions (a part of the whole brain), and the model is then evaluated by predicting the BOLD responses in the rest of the brain with correct decoded behaviors, driven by the these estimated stimulation inputs. In addition, Table 3 provides the comparison between the experimental paradigm of our mouse DTB model and traditional machine learning model.

## D STRUCTURE OF MOUSE DTB

We constructed the biologically plausible, weighted directed connectivity of the mouse whole brain at single-neuron resolution based on high-resolution axonal projection data (Oh et al., 2014) and spatial distribution of cells from mouse brain cell atlas (Zhang et al., 2023) through the proposed data-driven pipeline, as discussed in Section 3.1 and depicted in Figure 2. Referenced names and abbreviations of mouse brain regions are listed in Table 4.

Table 4: List of referenced mouse brain regions.

| Category | Brain regions |
|---|---|
| Major brain divisions | Cerebrum (CH), brainstem (BS), cerebellum (CB) |
| Major brain regions | Isocortex, olfactory areas (OLF), hippocampal formation (HPF), cortical subplate (CTXsp),striatum (STR), pallidum (PAL), thalamus (TH), hypothalamus (HY),midbrain (MB), pons (P), medulla (MY), cerebellum (CB) |
| Interoceptive regions | primary somatosensoty area (SSp), hippocampal region (HIP), amygdala (Amy), agranular insular area (AI), medial prefrontal cortex (mPFC), thalamus (TH), hypothalamus (HY), periaqueductal grey (PAG) |
| Olfactory regions | olfactory areas (OLF) |
| SSp related brain regions | motor area (MO), secondary somatosensory area (SSs), thalamus (TH) |
| HIP related brain regions | emporal association areas (TEa), retrohippocampal region (RHP), auditory area (AUD), SSp |
| OLF related brain regions | agranular insular area (AI), endopiriform nucleus (EP) |
| VIS related brain regions | the retrosplenial area (RSP), posterior parietal association areas (PTLp), TH |

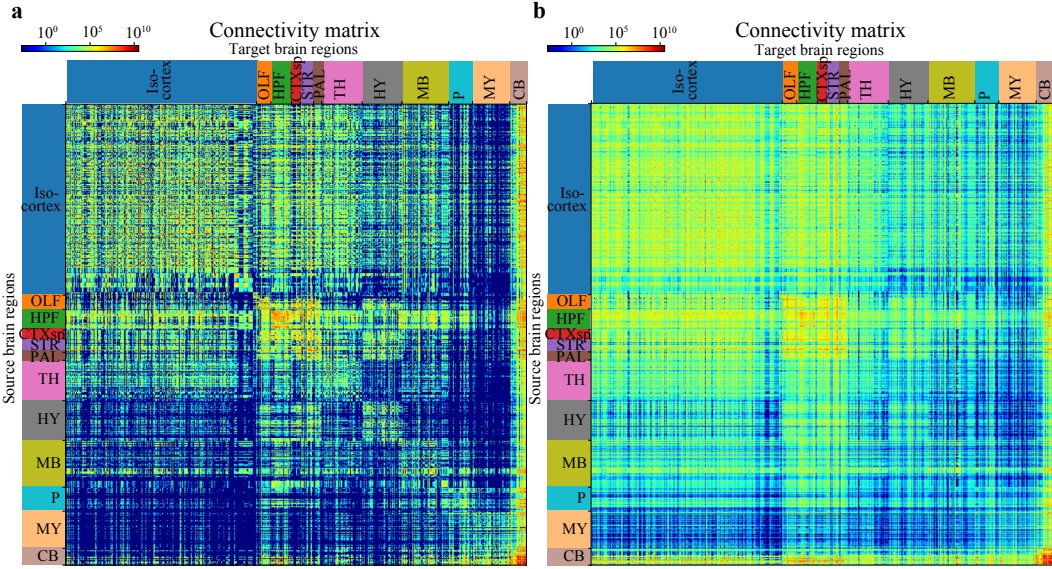

Figure 7: Connectivity between brain regions before and after iterative optimization. (a) Neuronal projection connectivity before iterative optimization. (b) Optimized connectivity.

### D.1 INFERENCE OF NEURONAL PROJECTION CONNECTIVITY

The initial neuronal projection connectivity is depicted in Figure 7a. Although the projection pattern of each injection experiment is normalized by its total injection density, no global normalization is applied across different experiments. Therefore, brain regions exhibit highly variable synaptic out-degrees, ranging from over 47 million neurons with no outgoing connections to a small subset of neurons with more than 4 million synaptic targets. As a result, the network became fragmented into several disconnected components, preventing coherent whole-brain communication. To resolve this inconsistency and promote a biologically plausible and regionally balanced projection architecture, it's necessary to iteratively optimize the neuronal projection connectivity matrix to enforce a more uniform distribution of neuronal out-degrees across brain regions.

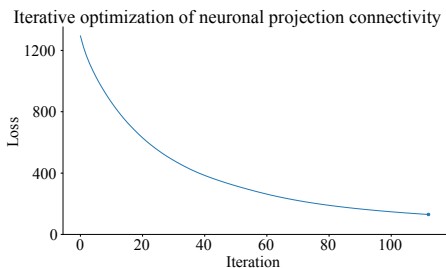

Figure 8: Loss curve of the iterative optimization of neuronal projection connectivity.

## D.2 ITERATIVE OPTIMIZATION OF NEURONAL PROJECTION CONNECTIVITY

To obtain a biologically plausible and regionally balanced projection structure, we performed iterative optimization on the neuronal projection connectivity matrix. The loss at iteration $t$ was defined as the L2 distance between the current neuronal out-degree $P_t$ and the target neuronal out-degree $Q = 16,000$, aggregated over all neurons. Accordingly, the relative loss change was computed as:

$$\Delta_{\text{loss}} = \frac{\text{Loss}_t - \text{Loss}_{t+1}}{\text{Loss}_t} \tag{14}$$

At iteration $t = 113$, the relative change in loss reached $\Delta_{\text{loss}} = 0.0099$, indicating that the optimization had reached a steady-state error of $1\%$, as depicted in Figure 8. We therefore generate the optimized mouse brain connectivity with 16,000 degrees by sampling connections based on the sampling probability of iteration 113. As shown in Figure 7b, the optimized mouse brain connectivity achieves balanced projection connectivity across brain regions, and ensures that the mouse whole brain network remains interconnected.

## D.3 GAUSSIAN LOCAL CONNECTIVITY

Due to the inherent limitations of axonal projection data, local connectivity within the injection sites are not represented in the axonal projection data. Thus, local connectivity is absent in the inferred neuronal projection connectivity. Therefor, we compensate for the missing local connectivity with additional Gaussian local connections.

To determine the appropriate proportion of local connectivity to be added, we conducted a set of interoception-driven resting-state simulations on mouse DTB models with varying synaptic degrees. These models were constructed using the optimized neuronal projection connectivity, but without any Gaussian local connections. As shown in Figure 9, the average correlation coefficient between the simulated and empirical BOLD signals across the whole brain increased steadily with synaptic degree. However, the improvement plateaued around a synaptic degree of 4,000, beyond which further increases did not yield significant gains. Given that the resting-state fMRI data mostly cover the cerebrum, the plateau at 4,000 degree suggests that approximately 50% of the synaptic degrees in the cerebrum's 8,000 degrees is sufficient to capture the functionally relevant mouse brain connectivity. Based on this finding, we replaced the other 50% of the neuronal projection connections in both the cerebrum and cerebellum with Gaussian local connections. However, in the brainstem, due to its significantly lower neuronal density, it's unable to sample enough Gaussian local connectivity. As a result, only 10% of the neuronal projection connections in the brainstem were replaced with Gaussian local connections to restore its local connectivity to the greatest extent possible.

By adding the Gaussian local connectivity, the cosine similarity for HPF neurons is 0.8834, significantly higher than 0.5973 for not adding local connectivity; while it is 0.6918 for PFC neurons, which is also significantly higher than 0.5611 for not adding local connectivity. Therefore, the single-neuron mouse brain connectivity with Gaussian local connectivity is consistent with independently measured single-neuron axonal projection data and is thus biologically plausible.

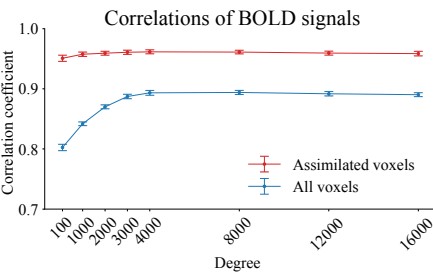

Figure 9: Effect of synaptic degree on interoception-driven resting state. Experiments are conducted on mouse DTB constructed with optimized connectivity and without Gaussian local connectivity.

Table 5: Details of the resting-state and olfactory Go/No-Go discrimination task fMRI data.

| T | TR (s) | Task | Brain regions | Voxels |
|---|---|---|---|---|
| | | Mouse DTB | All voxels | 8930 |
| 400 | 0.8 | Resting-state | Voxels with fMRI data | 6131 |
| | | | SSp, HIP, Amy, AI, mPFC, TH, HY and PAG | 2147 |
| 118 | 1.5 | Task-state | Voxels with fMRI data | 6800 |
| | | | OLF | 982 |
| | | | OLF, SSp, HIP | 2002 |
| | | | OLF, SSp, HIP, Amy, AI, mPFC, TH, HY and PAG | 3059 |
| | | | SSp, HIP, Amy, AI, mPFC, TH, HY and PAG | 2161 |

### D.4 ACTIVITY PROPAGATION ANALYSIS

1. Stimulation of the primary somatosensory area (SSp) leads to activation in the motor area (MO), secondary somatosensory area (SSs), and thalamus (TH), which are strongly connected to SSp.

2. Stimulation of the hippocampus (HIP) results in activity propagation to the temporal association areas (TEa) and retrohippocampal region (RHP) via direct connections, while auditory area (AUD) and SSp are likely activated through indirect pathways, demonstrating the ability of stimulation experiments on to reveal hidden multi-step connections.

3. Stimulation of the olfactory areas (OLF) elicits responses in the agranular insular area (AI) and endopiriform nucleus (EP), which are anatomically adjacent to and structurally connected with OLF.

4. Stimulation of the visual areas (VIS) activates the retrosplenial area (RSP), posterior parietal association areas (PTLp), and TH, which are involved in visual and visuospatial processing.

## E DETAILS OF THE FMRI DATA

The fMRI data used in this study were obtained and preprocessed following the procedures described in the original publication (Han et al., 2019). Details of the fMRI data are summarized in Table 5.

## F MOUSE DTB IN RESTING-STATE

### F.1 VALIDATION OF THE BIOLOGICAL PLAUSIBILITY OF THE ITERATIVE OPTIMIZATION.

The mouse DTB model without applying the iterative optimization discussed in Section 3.1.2 achieve a significantly lower correlation of 0.738 compared to the correlation of 0.901 for the baseline in the interoceptive-driven resting-state. This indicated that the iterative optimization is biolog-

Table 6: Resting-state correlation of randomly rewired mouse DTB.

| Rewiring percentages. | 0 | 0.2 | 0.4 | 0.6 | 0.8 | 1.0 |
|---|---|---|---|---|---|---|
| Stimulated regions corr. | 0.948 | 0.904 | 0.898 | 0.892 | 0.885 | 0.876 |
| Whole brain corr. | 0.901 | 0.729 | 0.699 | 0.659 | 0.606 | 0.527 |

ically plausible and necessary for simulating the resting-state dynamics. What's more, when comparing with the independent single-neuron axonal projection data (Qiu et al., 2024), by adding the Gaussian local connectivity, the cosine similarity for HPF neurons is 0.8834, significantly higher than 0.5973 for not adding local connectivity; while it is 0.6918 for PFC neurons, which is also significantly higher than 0.5611 for not adding local connectivity. Therefore, the single-neuron mouse brain connectivity with Gaussian local connectivity is consistent with independently measured single-neuron axonal projection data and is thus biologically plausible.

### F.2 VALIDATION OF SINGLE-NEURON CONNECTIVITY WITH INDEPENDENT AXONAL PROJECTION DATA.

We compared the cosine similarity of brain region level connectivity matrix between our single-neuron connectivity and the single-neuron axonal projections of 10,100 hippocampal (HPF) neurons and 6357 prefrontal cortex (PFC) neurons (Qiu et al., 2024). For HPF neurons, the cosine similarity of brain region level connectivity matrix (with 11 source regions across HPF and 30 target regions across the whole brain) is 0.8834. For PFC neurons, the cosine similarity reaches 0.6918. These results indicate that the projection patterns of key regions of our inferred single-neuron connectivity are consistent with independently measured single-neuron axonal projection data and are thus biologically plausible.

### F.3 COMPARISON WITH VOXEL-LEVEL MOUSE DTB.

We constructed a mouse DTB with a 400 $\mu$m voxel-level weighted connectivity of the mouse brain. The voxel-level mouse DTB shares the same spatial distribution of neurons and exc/inh neuron assignment with the single-neuron resolution mouse DTB. The voxel-level connectivity is inferred based on the axonal projections using the kernel regression approach in previous work (Knox et al., 2018), and governs the distribution of synaptic connections between voxels in the voxel-level mouse DTB. In each voxel, the in-coming connections are randomly distributed among all neurons in this voxel. In the interoceptive-driven resting-state, the voxel-level mouse DTB achieve a correlation of 0.819 across the whole brain, while our single-neuron mouse DTB achieve a significantly higher correlation of 0.901. This indicates that the single-neuron connectivity is not only consistent with the voxel-level connectivity constructed with well-established approach (Knox et al., 2018), but also enables simulation of mouse brain dynamics with higher correlation, due to its biological plausible heterogeneity of single-neuron connectivity.

### F.4 COMPARISON WITH RANDOMLY REWIRED OR LESIONED MOUSE DTB.

We tested the correlation of resting-state after randomly rewired a given percentage of synapses. As the percentage of rewired synapses increases, the connectivity becomes more arbitrary, and the correlation steadily drops from 0.901 to 0.527 (Table 6). We also ablated the synapses connecting to key brain regions, and tested the effect of individual brain region lesion on the correlation coefficients of resting-state (Table 7). Therefore, rewired or lesioned mouse DTB models exhibit a significant decrease in the correlation, demonstraing the vital role of the single-neuron connectivity for simulating whole brain dynamics in resting-state.

The details of random rewiring and lesion experiments:

1. **Data assimilation (re-training)**: We re-estimated the parameters governing the stimulation currents into the olfactory and key interoceptive regions, by fitting again the simulated BOLD signals to empirical BOLD signals in these regions using the HMDA method. As a result, driven by the estimated stimulation currents, the simulated and empirical BOLD

Table 7: Resting-state correlation of lesioned mouse DTB.

| Lesion region | Baseline | VIS | HPF | PFC | MO | TH | SSp |
|---|---|---|---|---|---|---|---|
| Number of neurons | - | 678k | 1849k | 717k | 1035k | 137k | 1277k |
| Stimulated regions corr. | 0.948 | 0.909 | 0.896 | 0.908 | 0.909 | 0.900 | 0.901 |
| Whole brain corr. | 0.901 | 0.737 | 0.678 | 0.715 | 0.694 | 0.743 | 0.656 |

signals in the stimulated regions still exhibit high correlation (close to the original score of 0.948), as shown by Corr. of the stimulation region in the following tables. We can assume that the stimulation input of olfactory signals and interoceptive signals is again faithfully reproduced in our rewired or lesioned mouse DTB model.

2. **Simulation (testing)**: Driven by the faithfully reproduced stimulation currents, the mouse DTB models with rewired or lesioned connectivity exhibited a significant loss in the correlation between the simulated (predicted) and empirical BOLD signals in the whole brain, demonstrating the significant role of the single-neuron connectivity in reproducing whole brain dynamics in resting-state.

## G  MOUSE DTB IN ACTION

### G.1  OLFACTORY TASK STATE

To investigate the influence of different brain regions in driving task-related dynamics, we evaluated the performance of the mouse DTB under four different input configurations during the olfactory discrimination task. As shown in Figure 10, the average Pearson correlation coefficient between simulated and empirical BOLD signals increased as more functionally relevant regions were incorporated as inputs.

When driven solely by the olfactory areas (OLF), the model achieved a relatively low correlation ( $0.247\pm0.018$), suggesting that OLF input alone was insufficient to reproduce the observed brain-wide dynamics. Adding the somatosensory and hippocampal inputs (OLF, SSp, HIP), which are previously proved to be significant for driving resting-state neural dynamics, substantially improved the correspondence ($0.554\pm0.019$), indicating that interoceptive regions also play a significant role in shaping task-related BOLD activity. The best performance was achieved when combining both olfactory sensory and interoceptive inputs (OLF + all interoceptive regions), reaching a correlation of $0.666\pm0.014$, which again highlights the importance of jointly modeling external stimuli and internal resting states for accurate task-state simulation.

### G.2  GO/NO-GO BEHAVIOR DECODING

To evaluate whether the simulated BOLD signals generated by the mouse DTB can support behaviorally meaningful readout, we applied a logistic regression decoder to classify Go versus No-Go responses.

To decode behavioral responses from BOLD signals, we adopted a **cross-validation** strategy across sessions. For each target session, the decoder was trained on empirical BOLD signals from the remaining sessions, and then applied to decode both the empirical BOLD data and the simulated BOLD data of the target session. This ensures that the decoding performance reflects generalizable stimulus-response mappings rather than session-specific overfitting.

As shown in Figure 11, behavior decoding performance was evaluated using three metrics: (a) **Accuracy**, defined as the overall proportion of correctly predicted trials (both Go and No-Go); (b) **Hit rate**, defined as the proportion of trials with odor 1 stimulation in which a correct Go response was predicted; (c) **Correct rejection rate**, defined as the proportion of trials with odor 2 stimulation in which a correct No-Go response was predicted.

Across all sessions, the mean decoding accuracy based on simulated BOLD signals reached $55.56\% \pm 9.39\%$. While this does not match the performance level achieved by decoding with empirical BOLD data, it is significantly above chance, suggesting that the simulated dynamics preserve

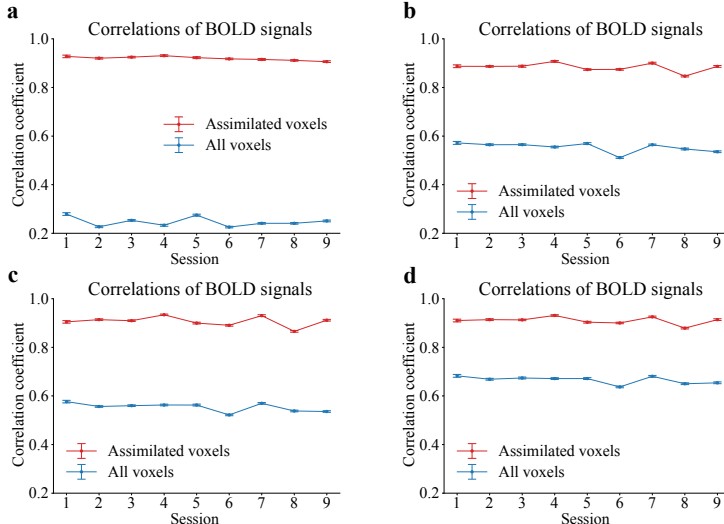

Figure 10: Correlation coefficients across all sessions driven by different set of brain regions. (a) Correlation coefficients driven by OLF. (b) Correlation coefficients driven by OLF, SSp and HIP. (c) Correlation coefficients driven by SSp, HIP, Amy, AI, mPFC, TH, HY and PAG. (d) Correlation coefficients driven by OLF, SSp, HIP, Amy, AI, mPFC, TH, HY and PAG.

task-relevant neural representations to a meaningful degree. The correct rejection rate achieved a mean of $87.17\% \pm 10.13\%$, which is comparable to performance levels observed in biological experiments, indicating that the DTB successfully captures the neural mechanisms underlying inhibitory responses. In contrast, the hit rate was lower, averaging $24.45\% \pm 20.14\%$, reflecting a limited ability of the DTB to consistently reproduce Go-related activation patterns. Nevertheless, in specific sessions (e.g., sessions 1, 6, and 9), the hit rate was substantially higher, indicating that the model can recover meaningful Go-responses under certain conditions.

Several factors may explain the limited performance observed when decoding behavior from simulated BOLD signals.

- First, the assimilation quality in certain perceptive regions may be insufficient. Some key ROIs within the assimilated region may not have achieved high correlation with the empirical BOLD, and important task-relevant regions outside the assimilated subset may also show poor alignment. These mismatches can limit the decoder's ability to extract consistent stimulus–response mappings from the simulated data.

- Second, although the simulated BOLD signals exhibit high voxel-wise Pearson correlation with empirical signals, this only reflects temporal similarity. In practice, the amplitude or baseline of simulated BOLD responses may differ due to multiplicative scaling factors or additive biases, which can adversely affect the decoder's output.

- Third, the decoder was trained on empirical data and may have overfitted to specific patterns present only in the experimental distribution. As a result, its generalization to DTB-generated data, which differ in scale, variability, or noise characteristics, may be impaired.

Simulation of Go/No-Go behavior under odor stimulation in the virtual rodent environment, including visualization of decoded behavior, neuronal spiking activity, and corresponding BOLD signals, is provided in the supplementary video.

# H    DISCUSSION

While our study presents a biologically grounded, single-neuron resolution digital model of the mouse brain, several limitations remain. The inferred connectivity relies on voxel-level projection

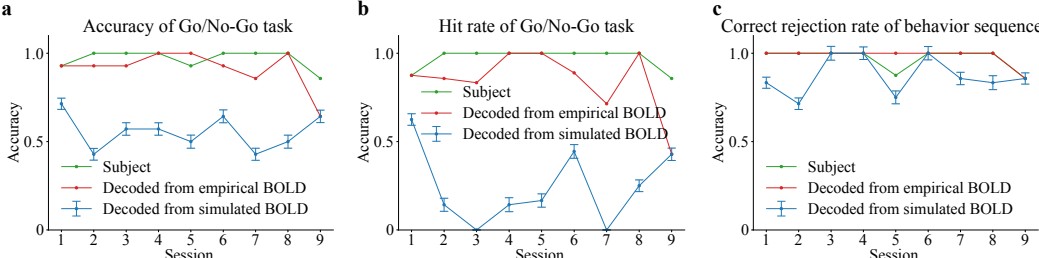

Figure 11: Performance of behavior decoded from simulated BOLD generated by mouse DTB. (a) Accuracy of decoded behavior. (b) Hit rate of decoded behavior. (c) Correct rejection rate of decoded behavior.

data and statistical approximation, which may not fully capture neuron-level or cell-type-specific variability. The spiking network model adopts simplified neuronal and synaptic dynamics. Future work will aim to incorporate plasticity, adaptive behavior, and more detailed physiological dynamics to further enhance the model's fidelity and utility for brain-wide functional simulation.

