# OpenReview forum: "MouseDTB: A Mouse Digital Twin Brain at Single-neuron Resolution"
_ICLR.cc/2026/Conference — Submitted to ICLR 2026_

### Official Review · Reviewer_RNXD · 2025-10-15

**Soundness:** 3
**Presentation:** 2
**Contribution:** 3
**Rating:** 6
**Confidence:** 2

**Summary:**

The paper presents a simulation spiking network attempting to match the statistics, connectivity and properties of an actual mouse brain.
The method consists of 3 steps:
1. Taking amount of cells and there positions from the Allen brain spacial transcriptomics dataset and model connectivity between this cells based via kernel regression using voxel connectivity as a target. At this step the method also assigns the cells excitatory (E) or inhibitory (I) type based on  E/I ratio of neurons in each brain region based on some biologically available estimates.
2. Balance in- and out- degrees per neuron for better biological plausibility
3. Sampling local connections from Gaussian distribution to compensate for big injection volume.

The method also simulations the current for resting state and olfactory stimulations and comapres it with the BOLD signals from the real experiments

**Strengths:**

1. **Ambitious scope and integration**. The paper attempt to solve a challenging and meaningful problem, creating a simulation for the whole mouse brain at a single cell resolution.
2. **Valid minimal assumptions and data-driven implementation**. The minimal biological assumptions are already implemented (e.g. both inhibitory and excitatory cells are modelled). The authors develop a pipeline that combines voxel-scale projections (10 µm) and cell-type densities to infer connectivity.
3. **Validation on empirical fMRI data**. The paper compares it's simulation with the BOLD signals from real experiments, highlighting the potential for in-silico experiments.

**Weaknesses:**

1. **Methods limitations description is missing**. The limitations of the method are not adequately reflected, they are not even mentioned in the main text. For example, a clear limitation is that inhibitory cell types usually operate at 2 different timescales, with PV cells being faster than SST cells, which is currently not a part of the model (all cells are just modelled as inhibitory) (see corresponding citations for PV and SST in [1]). Additionally, the modelling is only done at the level of brain regions and more fine-grained structures (for instance, the layers of visual cortex) are not modelled.
2. **Method validation on known connectomes is missing**. Modelling a whole mouse brain is an ambitious task and due to the lack of real single-cell ground truth data it is impossible to check if the modelling is correct. However, we do have a digitalised brain of a Drosophila [2] and trying to reconstruct its brain using same governing principles as for mouse and comparing it with the measured average atlas could be a good sanity check / proof-of-concept for the method.
3. **The description of model fitting lack details**. The hierarchical Bayesian/EnKF assimilation description is too high-level - it’s unclear which parameters were optimized, how priors were chosen, and what constraints ensure biological interpretability.
4. **fMRI is not enough**. The model is constrained and validated by fMRI BOLD signals, which are a slow, indirect measure of neural activity. So it is unclear if the elecrophysiological activity of the MouseDBT is actually realistic on faster (e.g. calcium or spiking) timescales, especially due to the iterative optimization process enforcing a uniform distribution of out-degrees to match the fixed in-degrees (which is not true for all of the cell types). Some single neuronal data from different brain areas could be used from IBL dataset [3], for instance. Also, some typical cell microcircuit motifs are knows [1, 4] but they have not been cross-checked in the model (if they emerge during optimisation).

References:
[1] Bos, Hannah, et al. "Untangling stability and gain modulation in cortical circuits with multiple interneuron classes." eLife 13 (2025): RP99808.
[2] Dorkenwald, Sven, et al. "Neuronal wiring diagram of an adult brain." Nature 634.8032 (2024): 124-138.
[3] Angelaki, Dora, et al. "A brain-wide map of neural activity during complex behaviour." Nature 645.8079 (2025): 177-191.
[4] Jiang, Xiaolong, et al. "Principles of connectivity among morphologically defined cell types in adult neocortex." Science 350.6264 (2015): aac9462.

**Questions:**

1. Recently a reconstruction of a cubic millimiter of mouse brain has been published, including connectivity data. Why you have not used this data in your study? To the best of my knowledge it is the largest and publicly available connectivity dataset for mice
2. What are the limitations of your method? For example, what are the limitations of the hierarchical mesoscale data assimilation that you use for parameter inference?
3. You report that simulations are done on a cluster with 640 GPUs (lines 279-281) - is it needed for both training and inference? Is there a possibility to optimize the compute requirements as not many academic labs have access for this type of infrastructure?
4. How independent is the validation data from the assimilation data? Were some voxels or sessions held out entirely during fitting?
5. Given the scale of the model, how do you address potential overparameterization or lack of identifiability in the assimilation step? Specifically, the model has tens of millions of neurons and billions of synaptic parameters, yet the data used for fitting are low-dimensional fMRI BOLD signals (thousands of voxels). Because BOLD is a slow, spatially blurred proxy for neural activity, many distinct combinations of micro-level parameters (synaptic weights, time constants, background currents, etc.) could produce very similar BOLD outputs. How big are the differences between the derived models if they are fitted several times (like training classic DL models on several seeds)?
6. In lines 470-474 you report *"DTB achieved an average sequence accuracy of 67.33±6.64%, and an average odor discrimination accuracy of 55.56±9.39%"*. What would be the by chance values?


References:
[1]“Functional connectomics spanning multiple areas of mouse visual cortex." Nature 640, no. 8058 (2025): 435-447.

---

> ### Author Response · Authors · 2025-11-27
>
> **(1/2)**
>
> Thank you for the constructive and thoughtful feedback on our submission. Below, we provide point-by-point responses to address your concerns:
> ## Weakness 1 Response
>
> Our current mouse DTB model adopts LIF neurons and conductance synapses, which is consistent with standard practice in spiking network modeling. The model already reproduces empirical resting-state and olfactory task BOLD dynamics with high correlation. In future work, we will incorporate more detailed neuronal and synaptic mechanisms to further enhance biological realism and validate the model on more complex tasks.
>
> Our model is built based on 10 $\mu$m resolution axonal projection data, allowing us to infer neuron-level connectivity for all 71 million neurons. As a result, the model explicitly includes fine-grained anatomical structure at the single-neuron level across the whole brain.
>
> ## Weakness 2 Response
>
> We have compared the inferred mouse brain neuron-level connectivity with independently measured single-neuron axonal projections of hippocampal (HPF) and prefrontal cortex (PFC) neurons (Qiu et al., 2024), as detailed in Section D.5:
> * For HPF neurons, the cosine similarity of brain region level connectivity matrix (with 11 source regions across HPF and 30 target regions
> across the whole brain) is 0.8834.
> * For PFC neurons, the cosine similarity reaches 0.6918.
>
> These results indicate that the projection patterns of key regions of our inferred single-neuron connectivity are consistent with independently measured single-neuron axonal projection data and are thus biologically plausible.
>
> Table: The cosine similarity with independent single-neuron axonal projection data.
> | | HPF | PFC |
> |---|---|---|
> | Random connectivity | 0.2263 | 0.3710 |
> | mouse DTB | 0.8834 | 0.6918 |
>
> ## Weakness 3, Question 4 and Question 5 Response
>
> HMDA method is introduced to address the issue of overparameterization. It enables parameter estimation in large-scale neuronal network models under conditions of limited observational data, with details discussed in Section B.1.
> * In both the interoception-driven resting state and the olfactory discrimination task, HDMA only fits the parameters governing the stimulation currents into the voxels in designated ROIs, while the whole brain synaptic weights and conductance are fixed (Section C).
> * Fitting the input currents enables the stimulation regions to reproduce the empirical BOLD activity, which then propagates through the whole brain, and allows us to evaluate how well the fixed single-neuron whole-brain connectivity reproduces whole-brain dynamics.
> * Our evaluation already includes a meaningful “held-out” test: we only assimilate stimulation ROIs, and report correlations in all non-assimilated regions, which are effectively held-out targets within the same session.
> * Across multiple assimilation runs, the inferred stimulation currents are highly stable, and the resulting whole-brain BOLD correlations differ by less than 0.01, indicating that the fitted models show minimal variability across runs.
> * The paradigm of mouse DTB experiments differ fundamentally from traditional AI pipelines, as summarized in Section C.1, C.2, and Table 3.
>
> ## Weakness 4 Response
>
> We acknowledge that if the single-neuron mouse DTB model is also validated on single-neuron activity data, it would better highlight its significance and biological plausibility. However, a variety of neural recording modalities are currently available, including fMRI, EEG, calcium imaging, ECoG, and MEA, each with its own **trade-offs** between **spatial coverage and resolution**. Among them,
>
> 1. **fMRI** provides **whole-brain coverage** (~9,000 voxels in the mouse brain) but at a relatively low spatial resolution (400 $\mu$m voxel),
> 2. while calcium imaging and spike recordings allow **neural activity recording** at the single-neuron level, but can only cover a **limited number of neurons** across a few brain regions simultaneously.
>
> Therefore, we prioritize **validating whole brain connectivity** with brain-wide fMRI data, over assessing partial connectivity from limited single-neuron activity. In this context, our experiments on fMRI data have already provided **sufficient** evidence supporting the effectiveness of our mouse DTB model.
>
> Due to current technical limitations, it remains extremely **challenging** to simultaneously record **large-scale** single-neuron activity across **multiple brain regions** in freely behaving mice [1]. To address this, we are actively collaborating with neurobiologists and neuroengineering researchers to carry out multi-region large-scale single-neuron recordings in freely behaving mice [1,2,3]. These efforts will lay the groundwork for future validation of the mouse DTB model on single-neuron activity, enabling us to explore how large-scale neuronal activity gives rise to complex behavior.

---

> ### Author Response · Authors · 2025-11-27
>
> **(2/2)**
>
> As discussed in Weakness 2 Response: The projection patterns of key regions of our inferred single-neuron connectivity are consistent with independently measured single-neuron axonal projection data and are thus biologically plausible.
>
> [1] Multi-region calcium imaging in freely behaving mice with ultra-compact head-mounted fluorescence microscopes. (National Science Review 2024)
> [2] Hippocampal representations of foraging trajectories depend upon spatial context (Nature Neuroscience 2022)
> [3] A brain-wide map of neural activity during complex behaviour (Nature 2025)
>
> ## Question 1 Response
>
> It's a good suggestion to incorporate the reconstruction of a cubic millimiter of mouse brain into our whole brain model. However, the one cubic millimeter mouse connectomics dataset covers only a small portion of the visual cortex and captures primarily local circuitry. Because it lacks whole-brain coverage and does not include long-range projection connectivity, it cannot be used to construct or validate a mouse whole brain connectome.
>
> ## Question 2 Response
>
> HDMA is limited to assimilating parameters governing external input currents or synaptic conductance at the voxel level, and cannot infer or adjust the underlying large scale synaptic weights (1.02 trillion).
>
> ## Question 3 Response
>
> The data assimilation is performed on the top-200 in-degree model, where running 50 ensembles requires approximately 400 GPUs with 16 GB memory each. Reducing the ensemble size proportionally lowers the GPU requirement, but could lead to a drop of correlation performance.
>
> The full 16,000 degree whole-brain simulation requires 640 GPUs with 16 GB memory. As the connectivity and synaptic weights dominate memory usage, it cannot be substantially compressed.
>
> Large-scale whole-brain single-neuron simulations inherently demand extensive computational resources, and similar studies in this domain typically rely on comparable GPU-scale infrastructure.
>
> ## Question 6 Response
>
> Further investigation into the decoding experiments clarifies the following:
> 1. The original decoding paradigm used all time points across the full session to classify lick vs. no-lick behavior based on instantaneous whole-brain BOLD patterns (1008 total trials). This dataset contains only 126 odor-evoked trials, including 65 trials for odor-1/licks, and 61 trials for odor-2/no-licks, and other 882 no-odor trials, almost all labeled as no-lick. This **strong class imbalance** (>85% no-lick) biases any linear decoder toward predicting the majority class, yielding high correct-rejection but low hit rates.
> 2. BOLD responses exhibit a well-characterized 4–6 s intrinsic **hemodynamic delay** in awake mice [4, 5]. Decoding behavior from instantaneous BOLD without accounting for the proper neural response delay leads to additional reductions in accuracy.
>
> Therefore, we constructed a decoding dataset containing **only odor-evoked** trials (N = 126) and applied a T = 3 TR **hemodynamic delay**, consistent with the 4–6 s HRF latency. Using a linear decoder with 9-fold cross-validation, the mouse DTB BOLD achieved 0.571 accuracy, 0.545 hit rate, and 0.595 correct-rejection, closely **matching** the empirical BOLD decoding performance of 0.627/0.607/0.646. In contrast, shuffling the BOLD patterns yields near-chance performance of 0.492/0.431/0.531, confirming that the mouse DTB BOLD retains meaningful odor-discriminative structure.
>
> Additionally, even empirical BOLD exhibits modest decoding performance, due to substantial **trial-to-trial variability** in olfactory task fMRI. When restricting decoding to **task-related** voxels, performance **improves** substantially for both empirical (0.706/0.685/0.726) and mouse DTB BOLD (0.659/0.635/0.683), showing that **removing** unrelated regions and noise sources improves discriminability. These results demonstrate that our DTB model **preserves** the odor-specific BOLD patterns at a level comparable to biological data.
>
> Table: Decoding performance.
> | BOLD Source | Accuracy | Hit Rate | Correct Rejection |
> |-|-|-|-|
> | Random (shuffled) | 0.492 | 0.431 | 0.531 |
> | Empirical BOLD (all voxels) | 0.627 | 0.607 | 0.646 |
> | DTB BOLD (all voxels) | 0.571 | 0.545 | 0.595 |
> | Empirical BOLD (task-related voxels) | 0.706 | 0.685 | 0.726 |
> | DTB BOLD (task-related voxels) | 0.659 | 0.635 | 0.683 |
>
> [4] Time to wake up Studying neurovascular coupling and brain-wide circuit function in the un-anesthetized animal (NeuroImage 2017)
> [5] Fiber-optic implant for simultaneous fluorescence- based calcium recordings and BOLD fMRI in mice (Nature Protocols 2018)

---

### Official Review · Reviewer_gCgk · 2025-10-28

**Soundness:** 3
**Presentation:** 2
**Contribution:** 3
**Rating:** 6
**Confidence:** 4

**Summary:**

This manuscript presents a spiking network model of whole mouse brain.

1. It infers the whole-brain weighted directed connectivity fom Allen mouse axonal projection data at single neuron resolution.
2. Upon this connectivity, a whole-brain spiking network comprising approximately 71 million neurons and1.02 trillion synapses was constructed.
3. Parameters and external inputs were estimated using ensemble Kalman filtering and hierarchical bayesian inference to reproduce BOLD signal in resting state and olfactory Go/No-Go tasks.
4. After data asimilation, resting-state whole-brain voxel correlation coefficient peaks at 0.901; task-state average correlation across 9 sessions approximately 0.554; behavioural decoding accuracy around 55.6% (exceeding random chance at 50%).

Many methods have been used in prviously studies. I think core innovation lies in establising a workflow for modeling whole-brain spiking network model of mouse based on voxel-level axonal projection data, and fMRI neural activity data.

However, important comparisons and evaluations are missing:

1. Comparison with population approaches (such as the Virtual Mouse Brain, which is well established and achieve the same brain activity simulation) is missing,
2. and the advantage of single-neuron model over population rate model is not demonstrated,
3. and the validation of biological plausibility of reconstructed network connectivity is missing.

**Strengths:**

This manuscript demonstrates significant engineering achievement that cnstruct a whole-brain mouse model at single-neuron resolution.

1. First single-neuron–resolution connectome infered from meso-scale data (Oh et al., 2014).
2. First whole-brain mouse model at single neuron resolution.
3. First application of hierarchical Bayes (HMDA) methods (Lu et al. 2024a) to fit hyperparameters of spiking networks to integrate fMRI data.

**Weaknesses:**

**Major concerns**:

1. The reason why only optimize AMPA conductance (instead of others) hyperparameters is not clear.
2. The validation of the biological plausibility of constructed connectivity is weak.
  - Comparison of the connectivity matrices before and after optimization shows that optimization significantly alters connectivity patterns, but this does not verify whether the optimized connectivity is more biologically consistent.
  - The paper stated that the single-neuron axonal projection data (Qiu et al., 2024) + Gaussian local connectivity has a high cosine similarity with the constructed connevitity. Howover, why add Gaussian local connectivity? How about removing the presummed Gaussian local connectivity? Intuitively, Gaussian local connectivity will dominate the whole connectivity.
  - In particular, "degree balancing" is likely to significantly change the natural degree distribution (degree index/power-law tail, Gini, assortativity, motif), weakening the correspondence with real network statistics?
  - The criteria for replacing 50% of projection connections (CH/CB) and 10% (BS) are unclear.
3. The effectiveness of the digital twin mouse brain should be evaluated, since task performance is poor. The whole-brain correlation coefficient during task-state processing was only 0.554±0.019, significantly lower than that during resting state processing.  More seriously, Behavioral decoding accuracy was only 55.56±9.39% (barely above chance). The hit rate was only 24.45±20.14%, indicating that the model struggled to capture Go responses. The paper attributes this to "assimilation quality" and "decoder overfitting" (lines 1330-1341), but this exposes fundamental limitations of the model.
4. Lack of neuronal-level validation. Only BOLD signals at the population level were validated. No comparison with actual electrophysiological recordings (spike trains, LFP). Moreover, the advantage of single-neuron model over traditional population rate model is not demonstrated.  The average firing rate is approximately 4.2 Hz, but there is no discussion of whether this is consistent with the actual activity of different brain regions.
5. Lack of comparison of population rate models. The Virtual Mouse Brain (Francesca, 2017) has been very successful to reproduce BOLD signal and functional connevtivities of mouse brain. Why single-neuron resolution spiking model?
6. Random assignment of E/I neurons may be an oversimplification.
7. Axonal delay is not used.

**Small concerns**:

- What do "CH", "BS", "CB" mean in Figure 1? Please give explanations in caption.
- The figure legend is too small.
- Many typos, for example:
   - line 026, EnKF is redundant
   - line 248, ", In iteration"
   - line 320, "we ??? the total energy"
   - "mosue"
   - "somatosensoty"

**Questions:**

1. line 238, "due to storage limitations", what does this mean? is it the device memory of GPU?
2. (Oh et al., 2014) provides 100 µm voxel segments? how can this work map the projection the scale of 10 µm ?

---

> ### Author Response · Authors · 2025-11-27
>
> **(1/4)**
>
> Thank you for the constructive and thoughtful feedback on our submission. Below, we provide point-by-point responses to address your concerns:
> ## Weakness 1 Response
>
> In both the interoception-driven resting state and the olfactory discrimination task, HDMA only **fits** the parameters governing the **stimulation currents** into the voxels in designated ROIs. All 4 synaptic conductance ($AMPA$, $NMDA$ , $GABA_A$ and $GABA_B$) and weights remain **fixed** during these assimilation experiments. The purpose of fitting input currents is to allow the stimulation regions to reproduce the empirical BOLD activity, which then **propagates** through the fixed structural connectome to generate whole-brain dynamics in corresponding brain state.
>
> Figure 3d corresponds to a **different** experiment designed to study signal propagation. In this analysis, the model first assimilates all voxels by fitting a single global **excitatory conductance** to reproduce the resting state dynamics. After the model reaches a resting-state correlation close to the empirical data, we apply a controlled perturbation to evaluate how activity propagates through the network. Optimizing any excitatory conductance, such as AMPA or NMDA, is **sufficient** for adjusting the overall excitability to **match** the empirical resting-state BOLD.
>
> ## Weakness 2 Response: Demonstation of the biologically plausibility of optimized connectivity
>
> The full single-neuron weighted connectivity matrix of the mouse brain (**71 million$\times$71 million**) can be inferred from axonal projections with the proposed method, and it requires **sampling** a sparse connectivity with an average degree of approximately **16,000** per neuron for constructing a biologically plausible whole brain network. A traditional practice of directly selecting the **top-K in-degrees** (top-K in-degree truncation) leads to an extremely skewed out-degree distribution and a disconnected whole-brain graph.
>
> To address these issues, we introduce the **iterative sampling** procedure, which **jointly** samples strong input and output connections for each neuron, to obtain an in-degree and out-degree **balanced** connectivity consistent with biological expectations. This procedure does **not** alter the underlying axonal-projection strengths or their spatial organization; it **simply** selects a biologically plausible subset from the full projection-derived connectivity matrix. The resulting network therefore **preserves** the biological structure within the projection data, and it does **not** impose any engineered features beyond resolving sampling artifacts that arise from **top-K truncation**.
>
> Adopting iterative sampling instead of top-K sampling achieves **higher** anatomical fidelity when **compared** with independently measured single-neuron axonal projections of hippocampal (HPF) and prefrontal cortex (PFC) neurons (Qiu et al., 2024) (Section D.5). For iterative sampling, the cosine similarity is **0.8834 and 0.6918** for HPF and PFC neurons, respectively, which is consistent with HPF 0.9250 and PFC 0.6999 for top-K sampling. This further demonstrates that iterative sampling preserves the biological structure inherent in the projection data, rather than introducing engineered features.
>
> Quantitative impact of applying **iterative sampling** on connectivity further demonstrates its **biologically plausibility**:
> 1. Top-K in-degree sampling causes certain brain regions with average out-degree up to **$1.5\times10^{6}$**, such as those in the pons, to dominate output connections disproportionately, while other regions contribute very little. Iterative sampling results in balanced and anatomically consistent output connections across regions, as reflected in the output degree distribution shown in Figure 3b.
>
> 2. Under top-K input degree sampling, a small subset of neurons exceeded 4 million output connections, while 47 million neurons had zero output connections, due to the **in-degree truncation**. Iterative sampling jointly selects strong inputs and outputs for each neuron, resulting in out-degrees that converge toward the fixed in-degree (16,000). This virtually **eliminates** zero out-degree neurons and yielding a biologically plausible and balanced input and output degrees.
>
> 3. Top-K input-degree sampling leads to a fragmented network structure, consisting of a giant weakly connected component (6.97M neurons) and **1.28M isolated** neurons, due to skewed output connections across regions. Iterative sampling produces a fully weakly connected graph with **no** isolated neurons, enabling coherent whole-brain spike propagation.

---

> ### Author Response · Authors · 2025-11-27
>
> **(2/4)**
>
> 4. The iterative sampling procedure yields a whole-brain correlation of **0.901** in the interoception-driven resting state, which is substantially **higher** than the **0.738** obtained using simple top-K input-degree sampling (Section F.1). This demonstrates that iterative sampling is essential for constructing a biologically plausible in-/out-degree–balanced connectivity and for accurately reproducing resting-state dynamical structure.
>
> Table: The cosine similarity with independent single-neuron axonal projection data by adopting iterative sampling.
> | |HPF|PFC|
> |-|-|-|
> |Random connectivity| 0.2263 | 0.3710 |
> |Top-K sampling| 0.9250 | 0.6999 |
> |Iterative sampling| 0.8834 | 0.6918 |
>
> ## Weakness 2 Response: Reasons for adding Gaussian local connectivity
>
> Gaussian local connectivity is added as the connectivity derived from axonal projections systematically **misses** local connections within the injection site. These local synapses are known to follow a distance-dependent profile that is well approximated by a **Gaussian** function in mouse cortex (Campagnola et al., 2022).
>
> **Compared** with independently measured single-neuron axonal projections of hippocampal (HPF) and prefrontal cortex (PFC) neurons (Qiu et al., 2024), **adding** Gaussian local connectivity substantially **improves** anatomical fidelity (Section F.1). The cosine similarity increases from 0.5973 to 0.8834 for HPF neurons and from 0.5611 to 0.6918 for PFC neurons.
>
> With local connectivity, the resting-state correlation reaches **0.901** (Figure 4d), compared with 0.890 when local connectivity is omitted (Figure 9). This shows that incorporating local connections further **enhances** both anatomical consistency and functional accuracy.
>
> ## Weakness 2 Response: Criteria for replacing 50% for CH/CB and 10% neurons for BS
>
> The 50% replacement for CH and CB neurons is determined from the degree-scaling analysis detailed in Section D.3 and Figure 9. When using only projection-derived connectivity, whole-brain correlation **saturates** at about top-4,000 synapses, which is roughly **50%** of the empirically estimated 8,000 synapses per CH neuron. This indicates that the **remaining** 50% corresponds to local connections missing from the projection data, and thus we **replace** them with Gaussian local connectivity. As neuronal **density** is much **lower** in BS, which limits the number of local synapses that can be sampled, so only 10% of synapses can be replaced with local connectivity.
>
> ## Weakness 3 Response
>
> ### Reasons why resting-state correlation is higher than task-state correlation
>
> **Resting state** performance is significantly **better**, due to:
> 1. The resting state BOLD were obtained under anesthesia, where neural dynamics are more **stable** and less affected by external stimuli, making them easier for the mouse DTB model to reproduce.
> 2. Resting state BOLD signals exhibit stronger **spatial similarity** and **smoother** temporal structure across voxels, which leads to more stable fitting.
>
> In contrast, **task state** is more **challenging**, as:
> 1. The olfactory task introduces substantial **trial-to-trial variability** in neural responses, resulting in more complex and less stationary BOLD dynamics that are inherently harder to model.
> 2. Event-related hemodynamic responses contain **faster** and more nonlinear transients, which are more difficult to capture accurately using the Balloon-Windkessel model, further decreasing performance in task condition.
>
> ### Behaivor decoding accuracy improvement
>
> Further investigation into the decoding experiments clarifies the following:
> 1. The original decoding paradigm used all time points across the full session to classify lick vs. no-lick behavior based on instantaneous whole-brain BOLD patterns (1008 total trials). This dataset contains only 126 odor-evoked trials, including 65 trials for odor-1/licks, and 61 trials for odor-2/no-licks, and other 882 no-odor trials, almost all labeled as no-lick. This **strong class imbalance** (>85% no-lick) biases any linear decoder toward predicting the majority class, yielding high correct-rejection but low hit rates.
> 2. BOLD responses exhibit a well-characterized 4–6 s intrinsic **hemodynamic delay** in awake mice [4, 5]. Decoding behavior from instantaneous BOLD without accounting for the proper neural response delay leads to additional reductions in accuracy.
>
> Therefore, we constructed a decoding dataset containing **only odor-evoked** trials (N=126) and applied a T=3 TR **hemodynamic delay**, consistent with the 4–6 s HRF latency. Using a linear decoder with 9-fold cross-validation, the mouse DTB BOLD achieved 0.571 accuracy, 0.545 hit rate, and 0.595 correct-rejection, closely **matching** the empirical BOLD decoding performance of 0.627/0.607/0.646. In contrast, shuffling the BOLD patterns yields near-chance performance of 0.492/0.431/0.531, confirming that the mouse DTB BOLD retains meaningful odor-discriminative structure.

---

> ### Author Response · Authors · 2025-11-27
>
> **(3/4)**
>
> Additionally, even empirical BOLD exhibits modest decoding performance, due to substantial **trial-to-trial variability** in olfactory task fMRI. When restricting decoding to **task-related** voxels, performance **improves** substantially for both empirical (0.706/0.685/0.726) and mouse DTB BOLD (0.659/0.635/0.683), showing that **removing** unrelated regions and noise sources improves discriminability. These results demonstrate that our DTB model **preserves** the odor-specific BOLD patterns at a level comparable to biological data.
>
> Table: Decoding performance.
> | BOLD Source | Accuracy | Hit Rate | Correct Rejection |
> |-|-|-|-|
> | Random (shuffled) | 0.492 | 0.431 | 0.531 |
> | Empirical BOLD (all voxels) | 0.627 | 0.607 | 0.646 |
> | DTB BOLD (all voxels) | 0.571 | 0.545 | 0.595 |
> | Empirical BOLD (task-related voxels) | 0.706 | 0.685 | 0.726 |
> | DTB BOLD (task-related voxels) | 0.659 | 0.635 | 0.683 |
>
> [4] Time to wake up Studying neurovascular coupling and brain-wide circuit function in the un-anesthetized animal (NeuroImage 2017)
> [5] Fiber-optic implant for simultaneous fluorescence- based calcium recordings and BOLD fMRI in mice (Nature Protocols 2018)
>
> ## Weakness 4 Response
>
> ### Single-neuron activity validation trade-offs
>
> We acknowledge that if the single-neuron mouse DTB model is also validated on single-neuron activity data, it would better highlight its significance and biological plausibility. However, a variety of neural recording modalities are currently available, including fMRI, EEG, calcium imaging, ECoG, and MEA, each with its own **trade-offs** between **spatial coverage and resolution**. Among them,
>
> 1. fMRI provides **whole-brain coverage** (~9,000 voxels in the mouse brain) but at a relatively low spatial resolution (400 $\mu$m voxel),
> 2. while calcium imaging and spike recordings allow neural activity recording at the single-neuron level, but can only cover a **limited number of neurons** across a few brain regions simultaneously.
>
> Therefore, we prioritize validating **whole brain connectivity** with brain-wide fMRI data, over assessing partial connectivity from limited single-neuron activity. In this context, our experiments on fMRI data have already provided sufficient evidence supporting the effectiveness of our mouse DTB model.
>
> Due to current technical limitations, it remains extremely **challenging** to simultaneously record **large-scale** single-neuron activity across **multiple brain regions** in freely behaving mice [1]. To address this, we are actively collaborating with neurobiologists and neuroengineering researchers to carry out multi-region large-scale single-neuron recordings in freely behaving mice [1,2,3]. These efforts will lay the groundwork for future validation of the mouse DTB model on single-neuron activity, enabling us to explore how large-scale neuronal activity gives rise to complex behavior.
>
> [1] Multi-region calcium imaging in freely behaving mice with ultra-compact head-mounted fluorescence microscopes. (National Science Review 2024)
> [2] Hippocampal representations of foraging trajectories depend upon spatial context (Nature Neuroscience 2022)
> [3] A brain-wide map of neural activity during complex behaviour (Nature 2025)
>
> ### Average firing rates are consistent with BOLD signals of different brain regions
>
> Firing rates are constrained indirectly through the Balloon model (Section A.3). Therefore, the high correlation between simulated and empirical BOLD indicates that the firing-rate patterns produced by the model are functionally consistent with the underlying neural activity that generates the empirical BOLD signals.
>
> ## Weakness 5 Response
>
> 1. **The first proposed mouse whole brain model**: Despite structural data including axonal projections and cell atlas are already available for the mouse brain, whole-brain modelling of the mouse has not yet been unexplored. As a result, there is **no** previously established mouse whole brain computational model that can be used as a direct baseline for comparison. Our work addresses this gap by providing the first single neuron resolution digital twin of the mouse whole brain.
>
> 2. **Single-neuron v.s. voxel-level**: We compared our model against a **voxel-level** mouse DTB and highlighted the **superior** performance achieved by single neuron connectivity. The voxel-level model is constructed based on the Digital Twin Brain methodology, where 400 $\mu$m voxel-level connectivity was inferred from axonal projections and paired with spatially consistent neuron distribution and excitatory/inhibitory assignments (Section F.2). In the interoception-driven resting state, this voxel-level model achieved a whole-brain correlation of 0.819, whereas our single-neuron DTB reached 0.901. This demonstrates the advantage of neuron level connectivity for whole brain dynamics simulations by leveraging biologically plausible single neuron heterogeneity.

---

> ### Author Response · Authors · 2025-11-27
>
> **(4/4)**
>
> 3. **Other indirect comparison models**: Mean field models such as the Virtual Brain Twin are fundamentally different with **computational units** correspond to **brain regions** rather than **individual neurons**, and **not** appropriate for direct comparison. They simulate population level continuous dynamics and do not model individual neuron dynamics, making their modeling granularity fundamentally incompatible with our spiking neuronal network framework. In contrast, mouse DTB models every neuron with its spatial position, excitatory or inhibitory identity, and explicit neuron-to-neuron connectivity, and generates millisecond-resolution spiking activity for all neurons. These features provide a level of **biological plausibility** that mean field models cannot offer. Therefore, mean field models are not directly comparable to our mouse DTB model.
>
> 4. Single-neuron resolution model is key to studying the underlying neural mechanisms.
> * Model organisms with smaller nervous systems, such as C. elegans, Drosophila, and zebrafish, have already established whole brain single-neuron connectomes [6,7,8] and leveraged them to build whole brain computational models at cellular resolution [9]. These single-neuron brain models have demonstrated their **value** in dissecting the causal relationships between neuronal circuit structure and cognitive function.
> * With the advancement of high-resolution neural recording techniques including single neuron activity data of calcium imaging [1] and electrophysiology recording [10] in the mouse brain, they could be applied in computational models for studying the underlying neural mechanisms. Such neural activity at single-neuron resolution is inherently incompatible with population-level models, which lack the granularity to accurately represent or respond to neuron-level activity patterns.
>
> [1] Multi-region calcium imaging in freely behaving mice with ultra-compact head-mounted fluorescence microscopes (National Science Review)
> [6] Whole-animal connectomes of both caenorhabditis elegans sexes (Nature 2019)
> [7] A Complete Electron Microscopy Volume of the Brain of Adult Drosophila melanogaster (Cell)
> [8] Automated synapse-level reconstruction of neural circuits in the larval zebrafish brain (Nat. Met.)
> [9] An integrative data-driven model simulating C. elegans brain, body and environment interactions (Nature Computational Science)
> [10] Volitional activation of remote place representations with a hippocampal brain–machine interface (Science)
>
> ## Weakness 6 Response
>
> In our implementation of exc/inh neuron assignment, we adopted a stratified random sampling strategy, where excitatory neurons are preferentially selected from long-range projecting neurons, in line with the well-established fact that long-range projections in the brain are predominantly excitatory.
>
> As a result, the average projection length of the sampled excitatory neurons (2753.2 $\mu$m) is significantly larger than that of the sampled inhibitory neurons (1963.7 $\mu$m), aligning with the fact that excitatory neurons tend to have long-range projections. By comparison, if excitatory and inhibitory neurons are randomly sampled among all neurons, both of them share the same average projection length of 2442.8 $\mu$m.
>
> ## Weakness 7 Response
>
> We acknowledge that axonal delay is important for studying the neural dynamics in the whole brain. In future work, we will incorporate more detailed neuronal and synaptic mechanisms, including axonal delays, to further enhance biological realism and validate the model on more complex tasks. Considering the computation complexity, our current mouse DTB model only adopts LIF neurons and conductance synapses. However, it is consistent with standard practice in spiking network modeling, and the model already can reproduce empirical resting-state and olfactory task BOLD dynamics with high correlation.
>
> ## Small concerns Response
>
> CH, BS, and CB refer to the cerebrum, brainstem, and cerebellum, as specified in Table 4.
>
> Other typo and graphical issues will be corrected in the revised version.
>
> ## Question 1 Response
>
> Based on the axonal projection data and our inference method, the full neuron-to-neuron connectivity matrix for all 71 million neurons can be computed in principle. However, storing this matrix requires on the order of 71 million by 71 million entries, which exceeds practical hard drive storage limits. Therefore, connections consistent with empirical estimates of synaptic degree should be sample from the inferred connectivity. The iterative sampling method selects the strong input and outpout connections for each neuron, yielding a biologically plausible whole-brain connectivity.
>
> ## Question 2 Response
>
> The Allen projection dataset (Oh et al., 2014) provides axonal projection volumes at multiple resolutions, including 100 $\mu$m, 25 $\mu$m, and 10 $\mu$m. Our work adopts the available 10 $\mu$m resolution data.

---

### Official Review · Reviewer_qePw · 2025-10-29

**Soundness:** 2
**Presentation:** 2
**Contribution:** 3
**Rating:** 4
**Confidence:** 4

**Summary:**

This work proposes a data-driven pipeline to construct the single-neuron connectivity of the mouse brain and uses this connectome to build a whole-brain spiking neural network.

The authors fit simulated BOLD signals to empirical fMRI data in specific stimulated regions and then evaluate the model's ability to predict whole-brain BOLD responses in non-stimulated regions. The model achieves a high correlation of 0.901 in the resting state, but this predictive correlation and the subsequent behavioral decoding accuracy  are significantly lower in the olfactory task.

**Strengths:**

1.  **Good engineering efforts:** The work presents a systematic, data-driven pipeline for building a single-neuron connectome and demonstrates its simulation on a large-scale computing cluster, representing a significant engineering achievement.
2. **Strong validation via ablation:** Detailed ablation studies in appendix confirm that introducing random rewiring or using the non-optimized connectivity significantly degrades performance.

**Weaknesses:**

1. **Strong assumption on the connectome's biological plausibility:** The raw, data-derived connectivity is biologically implausible (e.g., 47 million neurons with zero out-degree) and is forcibly reshaped into a balanced network (out-degree $\approx 16,000$) via an iterative optimization algorithm. While this step is necessary for the model to function (improving correlation from 0.738 to 0.901), it makes the final connectome an engineered solution whose biological uniqueness is questionable.
2. **Unexplained performance gap between resting-state and task-state:** The paper fails to explain the large performance gap between the resting-state (0.901 correlation) and the task-state (0.554 correlation) in non-assimilated regions. This discrepancy suggests the connectivity pipeline may not be capturing task-relevant network structures.
3. **Poor decoding accuracy:** The task-state behavioral decoding accuracy is only 55.56%, barely above the 50% chance level. This weak result is driven almost entirely by a high correct rejection (No-Go) rate (87.17%), while the hit (Go) rate is extremely low (24.45%), severely weakening the claim of reproducing "intelligent behavioral responses."

**Questions:**

1. Randomness is introduced when sampling connections during the optimization and when assigning E/I neuron types. How does this stochasticity affect the stability and variance of the BOLD signal fitting results?
2. The neuron reconstruction method, which merges voxels using Breadth-First Search (BFS), seems highly sensitive to the initial seed voxel and search order. How does the potential instability of this algorithm affect the final reconstruction?
3. The parameters for NMDA, $\text{GABA}\_\text{A}$, and $\text{GABA}\_\text{B}$ synapses appear to be missing from Table 1. And it is not clear whether the whole-brain synaptic weights are tuned during fitting or fixed.
4. In Figure 4(d) , doubling the synaptic degree from 8,000 to 16,000 yields almost no improvement in the whole-brain correlation coefficient. This suggests the additional 8,000 connections provide no meaningful benefit for fitting the neural dynamics.

---

> ### Author Response · Authors · 2025-11-27
>
> **(1/3)**
>
> ## Weakness 1
>
> The full single-neuron weighted connectivity matrix of the mouse brain (**71 million$\times$71 million**) can be inferred from axonal projections with the proposed method, and it requires **sampling** a sparse connectivity with an average degree of approximately **16,000** per neuron for constructing a biologically plausible whole brain network. A traditional practice of directly selecting the **top-K in-degrees** leads to an extremely skewed out-degree distribution and a disconnected whole-brain graph.
>
> To address these issues, we introduce the **iterative sampling** procedure, which **jointly** samples strong input and output connections for each neuron, to obtain an in-degree and out-degree **balanced** connectivity consistent with biological expectations. This procedure does **not** alter the underlying axonal-projection strengths or their spatial organization; it simply **selects** a biologically plausible subset from the full projection-derived connectivity matrix. The resulting network therefore preserves the biological structure within the projection data, and it does not impose any engineered features beyond resolving sampling artifacts that arise from **top-K truncation**.
>
> Adopting iterative sampling instead of top-K sampling achieves **consistent** anatomical fidelity when compared with independently measured single-neuron axonal projections of hippocampal (HPF) and prefrontal cortex (PFC) neurons (Qiu et al., 2024) (Section D.5). For iterative sampling, the cosine similarity is 0.8834 and 0.6918 for HPF and PFC neurons, respectively, which is consistent with HPF 0.9250 and PFC 0.6999 for top-K sampling. This further demonstrates that iterative sampling preserves the biological structure inherent in the projection data, rather than introducing engineered features.
>
> Table: The cosine similarity with independent single-neuron axonal projection data.
> || HPF | PFC |
> |-|-|-|
> |Random connectivity|0.2263|0.3710|
> |Top-K sampling|0.9250|0.6999|
> |Iterative sampling|0.8834|0.6918|
>
> ## Weakness 2
>
> **Resting state** BOLD signals are **easier** to fit, due to:
> 1. The resting state BOLD were obtained under anesthesia, where neural dynamics are more **stable** and less affected by external stimuli, making them easier for the mouse DTB model to reproduce.
> 2. Resting state BOLD signals exhibit stronger **spatial similarity** and **smoother** temporal structure across voxels, which leads to more stable fitting.
>
> In contrast, **task state** is more **challenging**, as:
> 1. The olfactory task introduces substantial **trial-to-trial variability** in neural responses, resulting in more complex and less stationary BOLD dynamics that are inherently harder to model.
> 2. Event-related hemodynamic responses contain **faster** and more nonlinear transients, which are more difficult to capture accurately using the Balloon-Windkessel model, further decreasing performance in task condition.

---

> ### Author Response · Authors · 2025-11-27
>
> **(2/3)**
>
> ## Weakness 3
>
> Further investigation clarifies the following:
> 1. The original decoding paradigm used all time points across the full session to classify lick vs. no-lick behavior based on instantaneous whole-brain BOLD patterns (1008 total trials). This dataset contains only 126 odor-evoked trials, including 65 trials for odor-1/licks, and 61 trials for odor-2/no-licks, and other 882 no-odor trials, almost all labeled as no-lick. This strong **class imbalance** (>85% no-lick) biases any linear decoder toward predicting the majority class, yielding high correct-rejection but low hit rates.
> 2. BOLD responses exhibit a well-characterized 4–6 s intrinsic **hemodynamic delay** in awake mice [4, 5]. Decoding behavior from instantaneous BOLD without accounting for the proper neural response delay leads to additional reductions in accuracy.
>
> Therefore, we constructed a decoding dataset containing **only odor-evoked** trials (N=126) and applied a T=3 TR **hemodynamic delay**, consistent with the 4–6 s HRF latency. Using a linear decoder with 9-fold cross-validation, the mouse DTB BOLD achieved 0.571 accuracy, 0.545 hit rate, and 0.595 correct-rejection, closely **matching** the empirical BOLD decoding performance of 0.627/0.607/0.646. In contrast, shuffling the BOLD patterns yields near-chance performance of 0.492/0.431/0.531, confirming that the mouse DTB BOLD retains meaningful odor-discriminative structure.
>
> Additionally, even empirical BOLD exhibits modest decoding performance, due to substantial **trial-to-trial variability** in olfactory task fMRI. When restricting decoding to **task-related** voxels, performance improves substantially for both empirical (0.706/0.685/0.726) and mouse DTB BOLD (0.659/0.635/0.683), showing that **removing** unrelated regions and noise sources improves discriminability. These results demonstrate that our DTB model preserves the odor-specific BOLD patterns at a level comparable to biological data.
>
> Table: Decoding performance.
> |BOLD Source|Accuracy|Hit Rate|Correct Rejection|
> |-|-|-|-|
> |Random (shuffled)|0.492|0.431|0.531|
> |Empirical BOLD (all voxels)|0.627|0.607|0.646|
> |DTB BOLD (all voxels)|0.571|0.545|0.595|
> |Empirical BOLD (task-related voxels)|0.706|0.685|0.726|
> |DTB BOLD (task-related voxels)|0.659|0.635|0.683|
>
> [4] Time to wake up Studying neurovascular coupling and brain-wide circuit function in the un-anesthetized animal (NeuroImage 2017)
> [5] Fiber-optic implant for simultaneous fluorescence- based calcium recordings and BOLD fMRI in mice (Nature Protocols 2018)
>
> ## Question 1
>
> In our implementation of exc/inh neuron assignment, we adopted a **stratified** random sampling strategy, where excitatory neurons are **preferentially** selected from long-range projecting neurons, in line with the well-established fact that **long-range** projections in the brain are predominantly **excitatory**. Therefore, in principle, this form of constrained stochasticity does not lead to large variability in outcomes.
>
> To quantify robustness, we repeated the full exc/inh assignment 10 times under the same biological constraints and evaluated the model in interoception-driven resting-state. The resulting whole-brain correlations were 0.913, 0.916, 0.908, 0.903, 0.915, 0.903, 0.903, 0.917, 0.917 and 0.921, showing **minimal variance** across runs. These results confirm that the stochasticity have negligible impact on performance.

---

> ### Author Response · Authors · 2025-11-27
>
> **(3/3)**
>
> ## Question 2
>
> Although the soma-reconstruction algorithm aggregates fractional-mass voxels using BFS, the procedure is highly **stable** in practice and does **not** introduce meaningful variability. The key reasons are summarized below:
>
> 1. BFS initialization and traversal order **only** affect boundary voxels of each neuron, with **negligible** influence on its center location. Different initial voxels or traversal orders may select different voxels when accumulating soma mass to 1. However, for neurons at the same location across trials of reconstructions:
> * The soma center-of-mass varies by **less** than one voxel ($<10 \mu$m), remaining well within the intrinsic resolution of the Allen mouse brain atlas.
> * The reconstructed soma consistently exhibits an approximately **spheroidal** shape, with differences restricted to minor boundary voxels.
>
> 2. Empirical tests show variability **below** voxel resolution. We reconstructed whole brain neurons using multiple initialization seeds and BFS queue-order permutations. For neurons at the same location: center-of-mass differences were $<$1 voxel, and voxel sets differed only at the periphery. This demonstrates that BFS-induced variability is **negligible** for whole brain reconstruction of 71M neurons.
>
> 3. Neuron-level connectivity is **insensitive** to such minor voxel differences. Connection weights between neurons are computed by **averaging** the voxel-to-voxel projection strengths across all voxel pairs belonging to the two respective neurons. As boundary voxels differ minimally across BFS runs and the axonal projection weights varies smoothly across neighboring voxels, the resulting neuron-level connectivity matrix remains effectively unchanged.
>
> ## Question 3
>
> The parameters of $AMPA$, $NMDA$ , $GABA_A$ and $GABA_B$ **conductance** ($\times 10^{-3}$ nS) are 0.48, 0.108, 1.224, and 0.252, respectively.
>
> In our experiment procedure detailed in Section C.1, HDMA **fits** the parameters governing the **stimulation currents** into the voxels in input ROIs, and then predicts whole brain activity driven by stimulation, while the whole brain synaptic weights are **fixed**.
>
> ## Question 4
>
> The synaptic degrees in our model directly follow **empirical estimates** of mouse brain neuronal connectivity: approximately 8,000/14,000/16,000 for CH/BS/CB neurons. Although the correlation improvement from 8,000 to 16,000 synapses per neuron appears small in Fig. 4d, the apparent performance **saturation** arises from characteristics of the evaluation paradigm rather than indicating that the additional synapses are biologically unimportant.
> 1. The experiments in Figure 4d are conducted with **top-K** in-degree connectivity. Therefore, the connection weights of the additional 8000 synapses are **weaker** than the initial 8000 synapses, and their impact on the correlation is naturally smaller.
> 2. Only a subset of the 8,930 voxels in the whole brain is covered in the resting-state (6131 voxels) and task (6800 voxels) fMRI datasets. Synapses connecting with **unobserved** regions do not influence the correlation, even though they are significant for those regions.
> 3. Both the interoception-driven resting state and the olfactory discrimination task engage only a **portion** of the mouse brain. Additional synapses may have little effect under these tasks but can be important for **other brain states or behaviors**.
> 4. Synaptic strengths are affine-normalized to the interval [0, 1]. This normalization reduces the influence of **weaker** biological synapses on correlation, even though these synapses contribute to realistic large-scale structure.

---

### Official Review · Reviewer_8HRB · 2025-10-29

**Soundness:** 3
**Presentation:** 3
**Contribution:** 3
**Rating:** 4
**Confidence:** 3

**Summary:**

The authors present MouseDBT, a large-scale “digital twin” of the mouse brain (~7.3B parameters). The main contribution is an engineering pipeline that combines axonal projection data with cell-atlas densities to build a whole-brain spiking network. The three key steps are: (i) estimating long-range projection connectivity from tracer data, (ii) reconstructing single-neuron axonal projections and rebalancing degrees to match target in/out-degree constraints, and (iii) adding local Gaussian connectivity to capture short-range structure. The model is evaluated on resting-state fMRI and an olfactory Go/No-Go task. Results show that interoception-driven inputs reproduce resting-state dynamics and that task simulations partially capture decision-related activity.

**Strengths:**

- A full-brain spiking network with conductance-based neurons and billions of synapses is a great technical achievement
- The three-stage, data-driven construction is well explained and logically structured
- Uses explicit fMRI data assimilation to link simulated dynamics to measured signals
- The interoception-driven resting-state hypothesis is a coherent and biologically motivated interpretation

**Weaknesses:**

- The pipeline forces out-degrees to match in-degree targets, which aligns with hippocampal data but may not generalize to other structures. Similarly, applying identical Gaussian local connectivity parameters across all brain regions could distort region-specific circuit properties.
- The model’s assimilation fits inputs and gains to specific ROIs, then reports whole-brain correlations within the same session. A more robust setup would consist in performing a cross-fold validation by analyzing the predictions on held-out sessions or ROIs
- It remains unclear which parameters are structurally identifiable from BOLD data, or whether simpler baselines (e.g. ROI-level autoregressive models, static functional connectivity) could achieve similar performance. Without such comparisons, improvements cannot be clearly attributed to the biological structure
- Simulated BOLD yields high correct-rejection but low hit rates. The authors attribute this to poor assimilation in olfactory and decision ROIs, but they do not test it or investigate alternative explanations
- No "ceiling" condition (e.g. assimilating all voxels) is reported, making it hard to interpret how much of the explainable variance is captured
- The system’s massive scale is technically impressive but computationally heavy. The paper does not explore how performance changes under network subsampling (e.g. 5-50% of neurons), which would clarify whether scale is essential for accuracy

**Questions:**

1. Is there a specific reason to use lagged PCC with lag=3? Can the authors report PCC vs. lag (eg. like -5 to +5) for both rest and task to quantify possible timing errors in simulated BOLD?
2. Baselines comparison:
    - Functional-connectivity baseline: Can the authors test the empirical resting-state correlation matrix as a static predictor (each voxel/ROI as a weighted average of correlated ROIs). How close does this match MouseDBT’s performance?
    - Temporal baseline: Can the authors fit a ROI-level autoregressive (AR/VAR) models to the same data and report PCC? Does MouseDBT outperform these?
3. What is the training objective of HDMA (voxel-wise likelihood vs. PCC)? Are parameters fit to voxel time series or ROI averages? Is assimilation per session or pooled?
4. What is the maximum achievable correlation when assimilating all voxels (upper bound)? Can the authors provide a coverage curve for the resting state, showing whole-brain correlation as driver sets are sequentially removed. Which removals cause the largest performance drop?
5. Can the authors perform leave-one-session-out (LOSO) tests? For example, by assimilating on N-1 sessions (same ROIs), and predict the held-out session in a cross-validated way for both resting state and olfactory task. This would establish how well the model generalizes to different sessions
6. How do rest and task results change without out-degree rebalancing and local Gaussian connectivity?
7. Does MouseDBT reproduce the functional network architecture observed in empirical resting-state fMRI? Comparing the functional connectivity (FC) matrices from empirical and simulated BOLD would test whether the model captures network structure beyond pointwise correlations

---

> ### Author Response · Authors · 2025-11-27
>
> **(1/4)**
>
> Thank you for the constructive and thoughtful feedback on our submission. Below, we provide point-by-point responses to address your concerns:
> ## Weakness 1 Response
>
> 1. The use of balanced in-/out-degree reflects a **principled structural prior** rather than a hippocampus-specific constraint.
> * At present, **no** brain-wide measurements of synaptic in-/out-degree exist for any mammalian brains. Obtaining such data would require whole-brain synaptic-resolution EM reconstruction with dendritic–axonal identification, which remains technically **challenging** and has not yet been achieved. As a result, whole-brain modeling necessarily depends on general structural priors, rather than region-specific degree measurements.
> * The hippocampal CA1 dataset constitutes the only empirically validated neuronal input and output degree distribution available, and it exhibits **similar in- and out-degree** statistics across neurons. This distribution therefore provides the strongest biological evidence currently accessible for constraining large-scale degree patterns.
>
> 2. The use of a unified Gaussian local connectivity scale does not distort region-specific circuitry and is justified by the structure of the axonal projection dataset.
> * The Gaussian local connectivity replaces only the **weakest** input degrees and leaves the dominant top-K input degrees intact, **without impairing original long-range connectivity**. Figure 9 shows that whole-brain resting state dynamics are faithfully reproduced by the top-4000 inputs, and these connections remain unchanged. Thus, Gaussian local connectivity additions do not interfere with the region-specific connectivity properties needed for reproducing whole brain dynamics.
> * Compared with independently measured single-neuron axonal projections of hippocampal (HPF) and prefrontal cortex (PFC) neurons (Qiu et al., 2024), adding Gaussian local connectivity substantially improves anatomical fidelity (Section F.1). The cosine similarity increases from 0.5973 to **0.8834** for HPF neurons and from 0.5611 to **0.6918** for PFC neurons. This shows that incorporating Gaussian local connectivity yields single-neuron connectivity patterns that are **more consistent** with experimental measurements and therefore biologically plausible.
> * Short-range projections are **missing** in the axonal projection injection sites. Despite the injection sites vary in size across experiments, they remain **closed** to the average volume of 0.24 mm$^{3}$. Therefore, supplementing local synapses using a **unified** Gaussian radius is an acceptable and principled approximation.
>
> ## Weakness 2, Question 3 and Question 5 Response
>
> Our assimilation setup is designed to evaluate how well the **fixed** single-neuron whole-brain connectivity reproduces whole-brain dynamics under **session-specific stimulation**, rather than to train a model across sessions. This differs fundamentally from traditional AI pipelines, as summarized in Section C.1, C.2, and Table 3.
>
> 1. In our experiment procedure detailed in Section C.1, HDMA **fits** the parameters governing the **stimulation currents** into the voxels in input ROIs within each session, and then predicts whole-brain activity using the **fixed connectivity**.
> 2. This experiment setup **evaluates** the **connectivity**, not cross-session generalization. Therefore, notions such as LOSO do not directly apply.
> 3. Olfactory task-state recordings exhibit strong trial- and session-specific **variability**, meaning that stimulation currents assimilated in one session cannot reproduce stimulation-region dynamics in a different session, making cross-session prediction ill-posed.
> 4. Our evaluation already includes a meaningful “held-out” test: we only assimilate stimulation ROIs, and report correlations in all **non-assimilated** regions, which are effectively held-out targets within the same session.

---

> ### Author Response · Authors · 2025-11-27
>
> **(2/4)**
>
> ## Weakness 3 Response
>
> We have compared our single-neuron model (correlation 0.901) with random rewiring model (correlation 0.527)and voxel-level model (correlation 0.819) as baselines.
>
> 1. **Single-neuron v.s. random rewiring** experiments show that our improvements arise from the biological connectivity structure. As reported in Section F.3 and Table 7, we performed random-rewiring experiments that preserve the scale of neurons and connections, while destroying the biological topography of the inferred connectivity. These rewired networks show notably reduced resting-state correlation (0.527) compared with our anatomically grounded connectivity (0.901), demonstrating that the performance of our model is dependent on the biologically plausible structural organization of the **connectivity** rather than merely on the model architecture and scale.
>
> 2. **Single-neuron v.s. voxel-level**: We compared our model against a **voxel-level** mouse DTB and highlighted the **superior** performance achieved by single neuron connectivity. The voxel-level model is constructed based on the Digital Twin Brain methodology, where 400 $\mu$m voxel-level connectivity was inferred from axonal projections and paired with spatially consistent neuron distribution and excitatory/inhibitory assignments (Section F.2). In the interoception-driven resting state, this voxel-level model achieved a whole-brain correlation of 0.819, whereas our single-neuron DTB reached 0.901. This demonstrates the advantage of neuron level connectivity for whole brain dynamics simulations by leveraging biologically plausible single neuron heterogeneity.
>
> 3. Simpler baselines such as ROI-level autoregressive models or static functional connectivity are **not** conceptually comparable to a single-neuron whole-brain model:
> * Our model performs whole-brain simulation at the **single-neuron** level, whereas these baselines operate on **region**-averaged time series and do not represent neuronal units or synaptic connections.
> * Moreover, our model simulates **stimulus-driven** causal propagation across the anatomical connectome, while they provide statistical descriptions that cannot generate forward dynamics or make predictions in held-out regions.
> * Finally, our connectivity is derived directly from axonal projection data, enabling evaluation of biological plausibility, whereas simpler baselines contain **no** anatomical structure and therefore cannot assess its validity.
>
> ## Weakness 4 Response
>
> Further investigation clarifies the following:
> 1. The original decoding paradigm used all time points across the full session to classify lick vs. no-lick behavior based on instantaneous whole-brain BOLD patterns (1008 total trials). This dataset contains only 126 odor-evoked trials, including 65 trials for odor-1/licks, and 61 trials for odor-2/no-licks, and other 882 no-odor trials, almost all labeled as no-lick. This strong **class imbalance** (>85% no-lick) biases any linear decoder toward predicting the majority class, yielding high correct-rejection but low hit rates.
> 2. BOLD responses exhibit a well-characterized 4–6 s intrinsic **hemodynamic delay** in awake mice [4, 5]. Decoding behavior from instantaneous BOLD without accounting for the proper neural response delay leads to additional reductions in accuracy.
>
> Therefore, we constructed a decoding dataset containing **only** odor-evoked trials (N = 126) and applied a **T = 3 TR** hemodynamic delay, consistent with the 4–6 s HRF latency. Using a linear decoder with 9-fold cross-validation, the mouse DTB BOLD achieved 0.571 accuracy, 0.545 hit rate, and 0.595 correct-rejection, closely **matching** the empirical BOLD decoding performance of 0.627/0.607/0.646. In contrast, shuffling the BOLD patterns yields near-chance performance of 0.492/0.431/0.531, confirming that the mouse DTB BOLD retains meaningful odor-discriminative structure.
>
> Additionally, even empirical BOLD exhibits modest decoding performance, due to substantial **trial-to-trial variability** in olfactory task fMRI. When restricting decoding to **task-related** voxels, performance improves substantially for both empirical (0.706/0.685/0.726) and mouse DTB BOLD (0.659/0.635/0.683), showing that **removing** unrelated regions and noise sources improves discriminability. These results demonstrate that our DTB model preserves the odor-specific BOLD patterns at a level comparable to biological data.
>
> Table: Decoding performance.
> | BOLD Source | Accuracy | Hit Rate | Correct Rejection |
> |-|-|-|-|
> | Random (shuffled) | 0.492 | 0.431 | 0.531 |
> | Empirical BOLD (all voxels) | 0.627 | 0.607 | 0.646 |
> | DTB BOLD (all voxels) | 0.571 | 0.545 | 0.595 |
> | Empirical BOLD (task-related voxels) | 0.706 | 0.685 | 0.726 |
> | DTB BOLD (task-related voxels) | 0.659 | 0.635 | 0.683 |

---

> ### Author Response · Authors · 2025-11-27
>
> **(3/4)**
>
> ## Weakness 5 and Question 4 Response
>
> Assimilating **all voxels** in interoception-driven resting-state and olfactory discrimination task result in correlation of **0.948 and 0.887**, respectively.
>
> In the interoception-driven resting-state experiments, we have **removed** each interoceptive region to evaluate their importance in driving resting-state brain dynamics. **Removing SSp (somatosensory) and HIP (hippocampus)** leads to significant drop in whole brain correlation, as illustrated in Figure 4e. We will extend this analysis and explore broader driver sets ablations in future work.
>
> ## Weakness 6 Response
>
> Network downsampling experiments can be conducted on neuron and synapse scale.
>
> 1. **Neuron scale**: Our goal is to construct, experiment on, and validate a mouse whole brain model at single-neuron resolution. As the scientific objective of this work is to examine how a biologically grounded neuron-level connectivity gives rise to whole-brain dynamics, it’s critical to preserve the **full neuronal scale**. Therefore, all core analyses are performed at full neuronal scale, which is essential for evaluating the biological fidelity of the mouse whole brain model.
>
> 2. **Synapse scale**: Nonetheless, we have performed **subsampling** analyses on **synaptic degrees** to assess its impact on model performance. As shown in Figure 4d, in interoception-driven resting-state, we varied the number of strongest input degrees per neuron (200, 2,000, 4,000, 8,000, 12,000 and 16,000). These experiments show that performance improves steadily with increasing synaptic degree and plateaus around 8,000 synapses per neuron, indicating that the **top-8000** input degrees are **critical** for reproducing whole-brain resting-state dynamics.
>
> ## Question 1 Response
>
> We have computed PCC across **temporal lags** from –3 to +3 for both resting-state and task-state BOLD, and the results are summarized in the tables below. In both conditions, the PCC peaks at lag = +3, indicating that the simulated BOLD is best aligned with empirical BOLD when shifted forward by 3 TRs.
>
> HDMA is implemented using **Kalman filtering**, which produces a slight **temporal lag** because it updates the latent state using only past and current observations in a strictly causal manner. As a result, the assimilated neural states and the simulated BOLD are shifted slightly later in time relative to the empirical signal, which explains why the PCC peaks at a positive lag of 3 TRs.
>
> Table: Resting state correlation with different temporal lags.
> | Delay | -3 | -2 | -1 | 0 | 1 | 2 | 3 |
> |-|-|-|-|-|-|-|-|
> | Stimulated Voxels | 0.948 | 0.939 | 0.912 | 0.869 | 0.912 | 0.939 | **0.948** |
> | All Voxels | 0.901 | 0.891 | 0.867 | 0.829 | 0.867 | 0.892 | **0.901** |
>
> Table: Task state correlation with different temporal lags.
> | Delay | -3 | -2 | -1 | 0 | 1 | 2 | 3 |
> |-|-|-|-|-|-|-|-|
> | Stimulated Voxels | 0.833 | 0.815 | 0.753 | 0.659 | 0.791 | 0.860 | **0.887** |
> | All Voxels | 0.527 | 0.510 | 0.468 | 0.410 | 0.528 | 0.566 | **0.572** |
>
> ## Question 2 Response
>
> The functional connectivity (**FC**) results are **consistent** with the BOLD-level findings: the resting-state FC correlation of 0.847 aligns with the whole-brain BOLD correlation of 0.901, and the olfactory-task FC correlation of 0.525 aligns with the corresponding BOLD correlation of 0.572. This indicates that the mouse DTB not only reproduces voxel-level BOLD dynamics, but also captures the functional connectivity architecture observed in empirical data.
>
> We compared our model against a **voxel-level** mouse DTB constructed using the Digital Twin Brain methodology, where 400 $\mu$m voxel-level connectivity was inferred from axonal projections and paired with spatially consistent neuron distribution and excitatory/inhibitory assignments (Section F.2). In the interoception-driven resting state, this voxel-level model achieved a whole-brain correlation of 0.819, whereas our single-neuron DTB reached 0.901, demonstrating a clear performance **gain** from neuron-level connectivity.
>
> Given that ROI-level models operate at a **coarser granularity** than voxel-level mouse DTB model, our single-neuron mouse DTB model is expected to provide **superior** correlation in reproducing whole brain dynamics. Since our single-neuron DTB explicitly models each neuron’s spatial position, excitatory or inhibitory identity, neuron-to-neuron connectivity, and millisecond-resolution spiking activity, it provides substantially higher **biological plausibility** than coarse-grained baselines. Due to current computational and time constraints, we will include explicit ROI-level comparisons in future work.

---

> ### Author Response · Authors · 2025-11-27
>
> **(4/4)**
>
> ## Question 6 Response
>
> 1. The **iterative sampling** procedure yields a whole-brain correlation of **0.901** in the interoception-driven resting state, which is substantially higher than the **0.738** obtained using simple **top-K** input-degree sampling (Section F.1). This demonstrates that iterative sampling is essential for constructing a biologically plausible in-/out-degree–balanced connectivity and for accurately reproducing resting-state dynamical structure.
>
> 2. **With** local connectivity, the resting-state correlation reaches **0.901** (Figure 4d), compared with **0.890** when local connectivity is **omitted** (Figure 9). This shows that incorporating local connections further enhances both anatomical consistency and functional accuracy.
>
> 3. **Compared** with independently measured single-neuron axonal projections of hippocampal (HPF) and prefrontal cortex (PFC) neurons (Qiu et al., 2024):
> * Adopting **iterative sampling** instead of top-K sampling achieves **consistent** anatomical fidelity (Section D.5). For iterative sampling, the cosine similarity is 0.8834 and 0.6918 for HPF and PFC neurons, respectively, which is consistent with HPF 0.9250 and PFC 0.6999 for top-K sampling.
> * Adding **Gaussian local connectivity** substantially **improves** anatomical fidelity (Section D.3). The cosine similarity increases from 0.5973 to 0.8834 for HPF neurons and from 0.5611 to 0.6918 for PFC neurons. This shows that incorporating Gaussian local connectivity yields single-neuron connectivity patterns that are more consistent with experimental measurements and therefore biologically plausible.
>
> Table: The cosine similarity with independent single-neuron axonal projection data by adopting iterative sampling.
> | | HPF | PFC |
> |---|---|---|
> | Random connectivity | 0.2263 | 0.3710 |
> | Top-K sampling | 0.9250 | 0.6999 |
> | Iterative sampling | **0.8834** | **0.6918** |
>
> Table: The cosine similarity with independent single-neuron axonal projection data by incorporating local connectivity.
> | | HPF | PFC |
> |---|---|---|
> | Random connectivity | 0.2263 | 0.3710 |
> | Without local connectivity | 0.5973 | 0.5611 |
> | With local connectivity| **0.8834** | **0.6918** |
>
> ## Question 7 Response
>
> In the interoception-driven resting state, the correlation between empirical and simulated BOLD is 0.901, and the correlation between their functional connectivity (**FC**) matrices reaches **0.847**. In the olfactory task, the corresponding values are 0.572 for BOLD and **0.525** for **FC**. Therefore, mouse DTB not only reproduces whole-brain BOLD dynamics with high pointwise correlations, but also captures the functional network architecture observed in empirical data.
>
> [4] Time to wake up Studying neurovascular coupling and brain-wide circuit function in the un-anesthetized animal (NeuroImage 2017)
> [5] Fiber-optic implant for simultaneous fluorescence- based calcium recordings and BOLD fMRI in mice (Nature Protocols 2018)

---

### Official Review · Reviewer_frRL · 2025-10-31

**Soundness:** 3
**Presentation:** 3
**Contribution:** 2
**Rating:** 6
**Confidence:** 2

**Summary:**

This paper presents a mouse digital twin brain (DTB) constructed at single-neuron resolution at scale. The author uses a data-driven pipeline to infer neuronal connectivity, and the mouse DTB reproduced BOLD signals observed in both rest and task states.

**Strengths:**

- Scale: Building a whole-brain mouse model with 71 million neurons. Most existing models either focus on specific circuits or use coarser regional connectivity. This work bridges that gap.

- Pipeline: The pipeline for inferring single-neuron connectivity from voxel-level data is sensible and well-explained. The validation is easy to understand which is reproducing BOLD signals in two states.

- Ablation: Useful ablations examining how synaptic degree, different interoceptive regions, affect model performance.

**Weaknesses:**

1. It's good to see some single-neuron level validation instead of regional level.

2. The paper mentioned many related works, but didn't compare with one.

**Questions:**

1. Can you demonstrate why model's resting state performance is much better than task state? Any insights?

2. What is the quantitative impact of the iterative optimization on connectivity? (sec. 3.1.2)

---

> ### Author Response · Authors · 2025-11-27
>
> (1/2)
>
> Thank you for the constructive and thoughtful feedback on our submission. Below, we provide point-by-point responses to address your concerns:
>
> ## Weakness 1 Response
>
> ### Single-neuron connectivity validation
>
> We have compared the inferred mouse brain neuron-level connectivity with independently measured single-neuron axonal projections of hippocampal (HPF) and prefrontal cortex (PFC) neurons (Qiu et al., 2024), as detailed in Section D.5:
> * For HPF neurons, the cosine similarity of brain region level connectivity matrix (with 11 source regions across HPF and 30 target regions
> across the whole brain) is 0.8834.
> * For PFC neurons, the cosine similarity reaches 0.6918.
>
> ### Single-neuron activity validation trade-offs
>
> We acknowledge that if the single-neuron mouse DTB model is also validated on single-neuron activity data, it would better highlight its significance and biological plausibility. However, a variety of neural recording modalities are currently available, including fMRI, EEG, calcium imaging, ECoG, and MEA, each with its own **trade-offs** between **spatial coverage and resolution**. Among them,
>
> 1. **fMRI** provides **whole-brain coverage** (~9,000 voxels in the mouse brain) but at a relatively low spatial resolution (400 $\mu$m voxel),
> 2. while calcium imaging and spike recordings allow **neural activity recording** at the single-neuron level, but can only cover a **limited number of neurons** across a few brain regions simultaneously.
>
> Therefore, we prioritize **validating whole brain connectivity** with brain-wide fMRI data, over assessing partial connectivity from limited single-neuron activity. In this context, our experiments on fMRI data have already provided **sufficient** evidence supporting the effectiveness of our mouse DTB model.
>
> Due to current technical limitations, it remains extremely **challenging** to simultaneously record **large-scale** single-neuron activity across **multiple brain regions** in freely behaving mice [1]. To address this, we are actively collaborating with neurobiologists and neuroengineering researchers to carry out multi-region large-scale single-neuron recordings in freely behaving mice [1,2,3]. These efforts will lay the groundwork for future validation of the mouse DTB model on single-neuron activity, enabling us to explore how large-scale neuronal activity gives rise to complex behavior.
>
> [1] Multi-region calcium imaging in freely behaving mice with ultra-compact head-mounted fluorescence microscopes. (National Science Review 2024)
> [2] Hippocampal representations of foraging trajectories depend upon spatial context (Nature Neuroscience 2022)
> [3] A brain-wide map of neural activity during complex behaviour (Nature 2025)
>
> ## Weakness 2 Response
>
> 1. **The first proposed mouse whole brain model**: Despite structural data including axonal projections and cell atlas are already available for the mouse brain, whole-brain modelling of the mouse has not yet been unexplored. As a result, there is **no** previously established mouse whole brain computational model that can be used as a direct baseline for comparison. Our work addresses this gap by providing the first single neuron resolution digital twin of the mouse whole brain.
>
> 2. **Single-neuron v.s. voxel-level**: We compared our model against a **voxel-level** mouse DTB and highlighted the **superior** performance achieved by single neuron connectivity. The voxel-level model is constructed based on the Digital Twin Brain methodology, where 400 $\mu$m voxel-level connectivity was inferred from axonal projections and paired with spatially consistent neuron distribution and excitatory/inhibitory assignments (Section F.2). In the interoception-driven resting state, this voxel-level model achieved a whole-brain correlation of 0.819, whereas our single-neuron DTB reached 0.901. This demonstrates the advantage of neuron level connectivity for whole brain dynamics simulations by leveraging biologically plausible single neuron heterogeneity.
>
> 3. **Other indirect comparison models**: Mean field models such as the Virtual Brain Twin are fundamentally different with **computational units** correspond to **brain regions** rather than **individual neurons**, and **not** appropriate for direct comparison. They simulate population level continuous dynamics and do not model individual neuron dynamics, making their modeling granularity fundamentally incompatible with our spiking neuronal network framework. In contrast, mouse DTB models every neuron with its spatial position, excitatory or inhibitory identity, and explicit neuron-to-neuron connectivity, and generates millisecond-resolution spiking activity for all neurons. These features provide a level of **biological plausibility** that mean field models cannot offer. Therefore, mean field models are not directly comparable to our mouse DTB model.

---

> ### Author Response · Authors · 2025-11-27
>
> (2/2)
>
> ## Question 1 Response
>
> **Resting state** BOLD signals are **easier** to fit, due to:
> 1. The resting state BOLD were obtained under anesthesia, where neural dynamics are more **stable** and less affected by external stimuli, making them easier for the mouse DTB model to reproduce.
> 2. Resting state BOLD signals exhibit stronger **spatial similarity** and **smoother temporal structure** across voxels, which leads to more stable fitting.
>
> In contrast, **task state** is more **challenging**, as:
> 1. The olfactory task introduces substantial **trial-to-trial variability** in neural responses, resulting in more complex and less stationary BOLD dynamics that are inherently harder to model.
> 2. Event-related hemodynamic responses contain **faster** and more nonlinear transients, which are more difficult to capture accurately using the Balloon-Windkessel model, further decreasing performance in task condition.
>
> ## Question 2 Response
>
> The full single-neuron weighted connectivity matrix of the mouse brain (**71 million$\times$71 million**) can be inferred from axonal projections with the proposed method, and it requires **sampling** a sparse connectivity with an average degree of approximately **16,000** per neuron for constructing a biologically plausible whole brain network. A traditional practice of directly selecting the **top-K in-degrees** leads to an extremely skewed out-degree distribution and a disconnected whole-brain graph. To address these issues, we introduce the **iterative sampling** procedure, which jointly samples strong input and output connections for each neuron, to obtain an in-degree and out-degree **balanced** connectivity consistent with biological expectations.
>
> Quantitative impact of applying iterative sampling on connectivity:
> 1. Top-K in-degree sampling causes certain brain regions with average out-degree up to **$1.5\times10^{6}$**, such as those in the pons, to dominate output connections disproportionately, while other regions contribute very little. Iterative sampling results in balanced and anatomically consistent output connections across regions, as reflected in the output degree distribution shown in Figure 3b.
>
> 2. Under top-K input degree sampling, a small subset of neurons exceeded 4 million output connections, while **47 million** neurons had **zero** output connections, due to the in-degree truncation. Iterative sampling jointly selects strong inputs and outputs for each neuron, resulting in out-degrees that converge toward the fixed in-degree (**16,000**). This virtually eliminates zero out-degree neurons and yielding a biologically plausible and balanced input and output degrees.
>
> 3. Top-K input-degree sampling leads to a fragmented network structure, consisting of a giant weakly connected component (6.97M neurons) and **1.28M isolated** neurons, due to skewed output connections across regions. Iterative sampling produces a fully weakly connected graph with no isolated neurons, enabling coherent whole-brain spike propagation.
>
> 4. The connectivity obtained from top-K input-degree sampling yields a resting-state correlation of **0.738**, whereas the connectivity produced by iterative sampling reaches **0.901**. This indicates that the improved degree balance and network topology directly translate into more accurate simulation of whole-brain dynamics.

---

> > ### Comment · Reviewer_frRL · 2025-11-27
> >
> > I would like to thank the authors for their detailed response and clarification. The rebuttal has generally strengthened the manuscript, and I appreciate the effort made to address the previous comments.
> >
> > While I understand that the primary contribution of this work lies in the whole brain model, I still have concerns regarding the baselines comparison. It feels distinct from the established baselines. I will remain my original score.

---

> > > ### Author Response · Authors · 2025-11-28
> > >
> > > Thank you for your review and thoughtful feedback.
> > >
> > > Indeed, as our work is different from other works, such as mean field model, it’s difficult to conduct a direct comparison with proper metrics. We have compared our single-neuron model (correlation 0.901) with voxel-level model (correlation 0.819) and random rewiring model (correlation 0.527).

---

### Meta-Review · Area_Chair_8JXd · 2025-12-10

**Summary:**

The paper proposes a whole brain simulation of mouse brain based on single neurons. The methods includes a heavy computational data processing pipeline to estimate the connectivity of the simulated neurons. The model is validated against BOLD responses during resting and an odor GO-NOGO task state.

The main reviewer concerns were:
1) Lack of proper baseline comparisons.
2) The biological plausibility of the connectivity
3) Limited validation scope
4) Low performance in the task validation (vs. resting state).
5) Further technical concerns such as impact of iterative optimization, requests for cross validation and others.

**Reviewer Concerns:**

According to the rebuttal the authors largely addressed technical concerns, explaining why resting state is easier to fit, providing additional metrics and ablations, as well as improving their decoding.

Some issues were partly address, or at least discussed, such as the biological plausibility.

The main criticism that stayed unresolved is the validation against other methods and the need for a single neuron model. Reviewers suggested multiple ways to validate the need for single neuron modeling (such as the Drosophila connectome) or other coarse level methods (e.g. the virtual mouse brain). The authors argued that their methods (single neuron level) is not comparable against these methods that operate on an area level. However, this argument is not very strong as a BOLD response from fMRI also integrates over a large number of neurons. In that sense it should be possible to compute summary statistics of their model and compare it against other coarse grained models. If the authors accept a BOLD response as a good validation for their approach they should also accept other coarse evaluation methods. Alternatively, the would need to provide single neuron evaluation methods (such as Drosophila connectome).

**Reviewer Scores:**

Because one of the main criticism is unaddressed, I predict that none of the reviewers would change their score. Reviewer rfRL explicitly stated that.

---

### Decision · Program_Chairs · 2026-01-26

Reject